# Detecting the effect of genetic diversity on brain composition in an Alzheimer's disease mouse model
Brianna Gurdon [1,2,7], Sharon C. Yates [3,7], Gergely Csucs [3], Nicolaas E. Groeneboom[3], Niran Hadad[1,6], Maria Telpoukhovskaia [1], Andrew Ouellette [1,2], Tionna Ouellette[1,4], Kristen M. S. O'Connell[1,2,4], Surjeet Singh [1,5], Thomas J. Murdy [1], Erin Merchant[1], Ingvild Bjerke [3], Heidi Kleven [3], Ulrike Schlegel [3], Trygve B. Leergaard [3], Maja A. Puchades [3], Jan G. Bjaalie [3,8] ✉ & Catherine C. Kaczorowski [2,4,5,8] ✉

Alzheimer's disease (AD) is broadly characterized by neurodegeneration, pathology accumulation, and cognitive decline. There is considerable variation in the progression of clinical symptoms and pathology in humans, highlighting the importance of genetic diversity in the study of AD. To address this, we analyze cell composition and amyloid-beta deposition of 6- and 14-month-old AD-BXD mouse brains. We utilize the analytical QUINT workflow- a suite of software designed to support atlas-based quantification, which we expand to deliver a highly effective method for registering and quantifying cell and pathology changes in diverse disease models. In applying the expanded QUINT workflow, we quantify near-global age-related increases in microglia, astrocytes, and amyloid-beta, and we identify strain-specific regional variation in neuron load. To understand how individual differences in cell composition affect the interpretation of bulk gene expression in AD, we combine hippocampal immunohistochemistry analyses with bulk RNA-sequencing data. This approach allows us to categorize genes whose expression changes in response to AD in a cell and/or pathology load-dependent manner. Ultimately, our study demonstrates the use of the QUINT workflow to standardize the quantification of immunohistochemistry data in diverse mice, - providing valuable insights into regional variation in cellular load and amyloid deposition in the AD-BXD model.

Alzheimer's disease (AD) is a multifaceted neurodegenerative condition broadly characterized by the accumulation of amyloid-beta plaques, neurofibrillary tangles, severe gliosis, and progressive neurodegeneration, leading to clinical symptoms and cognitive decline[1]. There is significant variation in the age at symptom onset and severity of cognitive decline; highly susceptible individuals exhibit early onset and rapid decline, while resilient individuals remain cognitively intact late in life and may display differences in neuropathological load[2–4]. Additional characterization of pathology development such as neurodegeneration, amyloid-beta deposition, and neuroinflammation is necessary to enhance our understanding of

the influence of this variation on clinical outcomes. This characterization is particularly crucial as alterations in brain tissue composition and the onset of neuropathology can precede, and possibly predict clinical symptoms. Thus, it serves as a valuable resource for defining disease subtypes and potential mechanisms of resilience[5–7].

Mouse models of AD offer the opportunity to study changes in brain pathology in a controlled manner to gain a better understanding of how AD manifests in humans[8]. To counteract the lack of heterogeneity in traditional inbred AD mouse models, we used the AD-BXD mouse population that better recapitulates the heterogeneity of genetic, molecular, and cognitive

[1]The Jackson Laboratory, Bar Harbor, ME, USA. [2]The University of Maine Graduate School of Biomedical Sciences and Engineering, Orono, ME, USA. [3]Neural Systems Laboratory, Institute of Basic Medical Sciences, University of Oslo, Oslo, Norway. [4]Tufts University Graduate School of Biomedical Sciences, Medford, MA, USA. [5]Department of Neurology, University of Michigan, Ann Arbor, MI, USA. [6]Present address: Translational Genomics Research Institute, Phoenix, AZ, USA [7]These authors contributed equally: Brianna Gurdon, Sharon C. Yates. [8]These authors jointly supervised this work: Jan G. Bjaalie, Catherine C. Kaczorowski. ✉e-mail: j.g.bjaalie@medisin.uio.no; kaczoro@med.umich.edu

features of human aging and AD[9]. The AD-BXD population was generated by crossing the 5XFAD AD mouse model on a congenic C57BL/6 J (B6) background with strains from the BXD panel[9]. Despite carrying the same five highly penetrant mutations associated with early-onset AD, the genetically diverse AD-BXD strains exhibit phenotypes across a continuum of severity, recapitulating the clinical and pathological variation of late-onset AD[9–12]. Since the relationship between symptomatology and changes in the composition of brain tissue is not fully understood, assessing changes in cell and pathology organization across a population that models the heterogeneity of human AD may highlight brain regions and cell types associated with AD-related decline[13,14].

In addition to characterizing the impact of AD on cell composition in mouse models, AD-related changes can be described by deviations in gene expression among different cell types of the brain. Bulk RNA-sequencing (RNAseq) is a common method to study gene expression profiles of brain regions of interest; however, gene expression data generated from tissue using this method reflects an average gene expression profile across heterogeneous cellular populations[15]. AD induces detrimental changes in brain composition[16]; therefore, changes in gene expression in bulk tissue may be masked or confounded by changes in cell-type composition across varying disease stages[17]. In many studies using RNAseq to determine AD disease signatures, it is unclear whether observed differences in gene expression among AD samples or between AD samples and controls are due to changes in transcriptional regulation or the relative proportions of different cell types in the tissue[18]. Considering the contribution of cell abundance when associating gene expression to disease traits is important for reducing spurious associations between AD phenotypes and gene expression[19–21]. Deconvolution methods have been created in an attempt to remedy this issue[22]; however, the performance of deconvolution tools is highly variable[19,23].

Ultimately, immunohistochemistry (IHC) quantification is the gold standard for measuring brain composition in mouse models of disease. When combined with brain-wide analysis methods that utilize reference atlases of the brain[24,25], IHC is a powerful tool that can be used to better understand changes in brain composition that occur with age and AD and the relative relationship between cellular load (percent-stain-positive coverage/per region area) and gene expression. The QUINT workflow[26], an open-source analysis solution for brain histology, combines tools for registering brain section images to a reference brain atlas (DeepSlice[27] and QuickNII[28]) with tools for extracting (ilastik[29]) and quantifying IHC-stained features (Nutil[30]). A key step in the QUINT workflow is that customized atlas-planes, derived from a three-dimensional reference atlas, are linearly registered to brain section images[28]; however, with morphological differences seen among mouse strains, disease states, and ages[31–35], and morphological distortions occurring during histological processing, linear registration is often insufficient to achieve accurate anatomical registration. Here, we add increased functionality to the QUINT workflow to improve the quality of the atlas-registration via nonlinear refinement (VisuAlign); and provide a means to verify the atlas-registration by systematic random sampling (QCAlign). We use this expanded workflow to characterize the regional composition of neurons, astrocytes, microglia, and amyloid-beta across the brains of the AD-BXD mice (including the AD-BXD founders: B6xB6:5XFAD and B6:5XFADxDBA/2J) at different ages, in regions defined by the Allen Mouse Brain Common Coordinate Framework v3 (CCFv3)[36]. We provide an expansive brain-wide characterization of diverse 5XFAD mice and (1) assess changes in cell and amyloid-beta composition between adult and middle-aged AD-BXD animals, (2) assess variation in cellular abundance among AD-BXD strains, and (3) interpret bulk RNAseq data with respect to cellular-abundance to differentiate effects driven by age with AD from effects driven by cellular composition in the hippocampal formation.

## Results
### Additional functionality added to the QUINT workflow supports high-throughput analysis of diverse AD-BXD strains
The original QUINT workflow was designed to support the quantification of IHC-stained (or other labeled) features in serial brain sections by linear

registration to a reference brain atlas in combination with feature extraction by supervised machine learning[26]. While this method works well for high-quality serial sections, IHC processing often leads to distortions, tears, and artifacts that impact the quality of the atlas-registration. Furthermore, since the reference atlases are based on standard reference animals (young adult male B6 mice in the case of the CCFv3)[36], sections originating from animals of varying ages and/or genetically diverse strains may also have anatomical differences relative to the reference template. Recognizing the need to customize linear atlas-registration and provide a better match of the atlas overlay on individual sections, an additional tool that supports nonlinear refinement was created and incorporated into the workflow (VisuAlign) (Fig. 1a), along with a quality control tool that utilizes systematic random sampling to validate the atlas-registration to each region (QCAlign) (Fig. 1a). Here we demonstrate the effectiveness of the expanded QUINT workflow to quantify diverse cellular and pathological features in AD-BXD mice (Fig. 1b) by quantifying all nuclei (thionine), neurons (NeuN), microglia (Iba1), astrocytes (GFAP) and amyloid-beta (AB1-42) in customized regions compiled from CCFv3 regions (Fig. 1c, d).

### Quality of the atlas-registration performed in the QUINT workflow can be confirmed using QCAlign
QCAlign works by positioning a systematic random sampling grid over brain sections that have been registered to a standard atlas using QuickNII and VisuAlign (Fig. 2a), allowing raters to assess whether the points have been registered correctly to their designated atlas region using anatomical expertise (Fig. 2b). Assigned marker counts (accurate, inaccurate, or uncertain) are used to calculate regional registration accuracy. Because only a limited number of landmarks can be revealed by IHC or other staining, the granularity of the reference atlas can be adjusted in QCAlign to a level that supports this validation. This provides users a platform for flexible assessment since individual reference regions can be compiled into larger themed regions (e.g., isocortex), allowing an assessment tailored to each unique experimental design. For this QCAlign assessment, the full CCFv3 2015 was condensed into 77 regions of interest (Supplementary Data 1), creating an intermediate hierarchy of regions that had visually discernable boundaries as detected in the thionine-stained sections (displayed for one brain series in Supplementary Fig. 1). Here QCAlign was implemented by multiple raters to assess the quality of the atlas-registration achieved using QuickNII (for linear registration) and VisuAlign (for nonlinear adjustment) in conjunction.

There was high a consensus among the raters that the intermediate hierarchy regions assessed after completing nonlinear adjustments with VisuAlign were registered to the brain sections with high accuracy (78.7–100% accuracy score) (Fig. 3a, green, Supplementary Fig. 2). Regions with the greatest accuracy scores were those compiled of many subregions (e.g., isocortex, 99.7 ± 0.057%) and/or those that had very distinct anatomical borders (e.g., caudoputamen, 99.4 ± 0.129%). Due to the selected sampling rate (15-voxel grid spacing), some smaller regions had zero grid markers placed within their area, reducing the number of assessments contributing to the calculation of the mean accuracy scores (e.g., subparafascicular area, $n = 9$ assessments) (Fig. 3a, green, see Supplementary Data 2 for the number of assessments measured per region). Regions with the lowest number of assessments were among the regions with the highest variation and lowest accuracy scores. Regions with appropriate rater sampling ($n > 20$ assessments) but lower accuracy scores included the posterior amygdalar nucleus (89.1% ± 5.37%) and the ventricular systems (78.7 ± 3.11%). The lower accuracy attributed to the posterior amygdalar nucleus could be due to its relatively ambiguous border with the posterior olfactory area and the subiculum. Regions of the ventricular system were consistently difficult to align in both QuickNII and VisuAlign since they are prone to distortion (e.g. lateral ventricle) or are located in medial locations along the midline where the brain was bisected (e.g. third ventricle), resulting in low accuracy overall. In summary, we created an additional tool for quality assessment of the atlas-registration and were able to confirm the

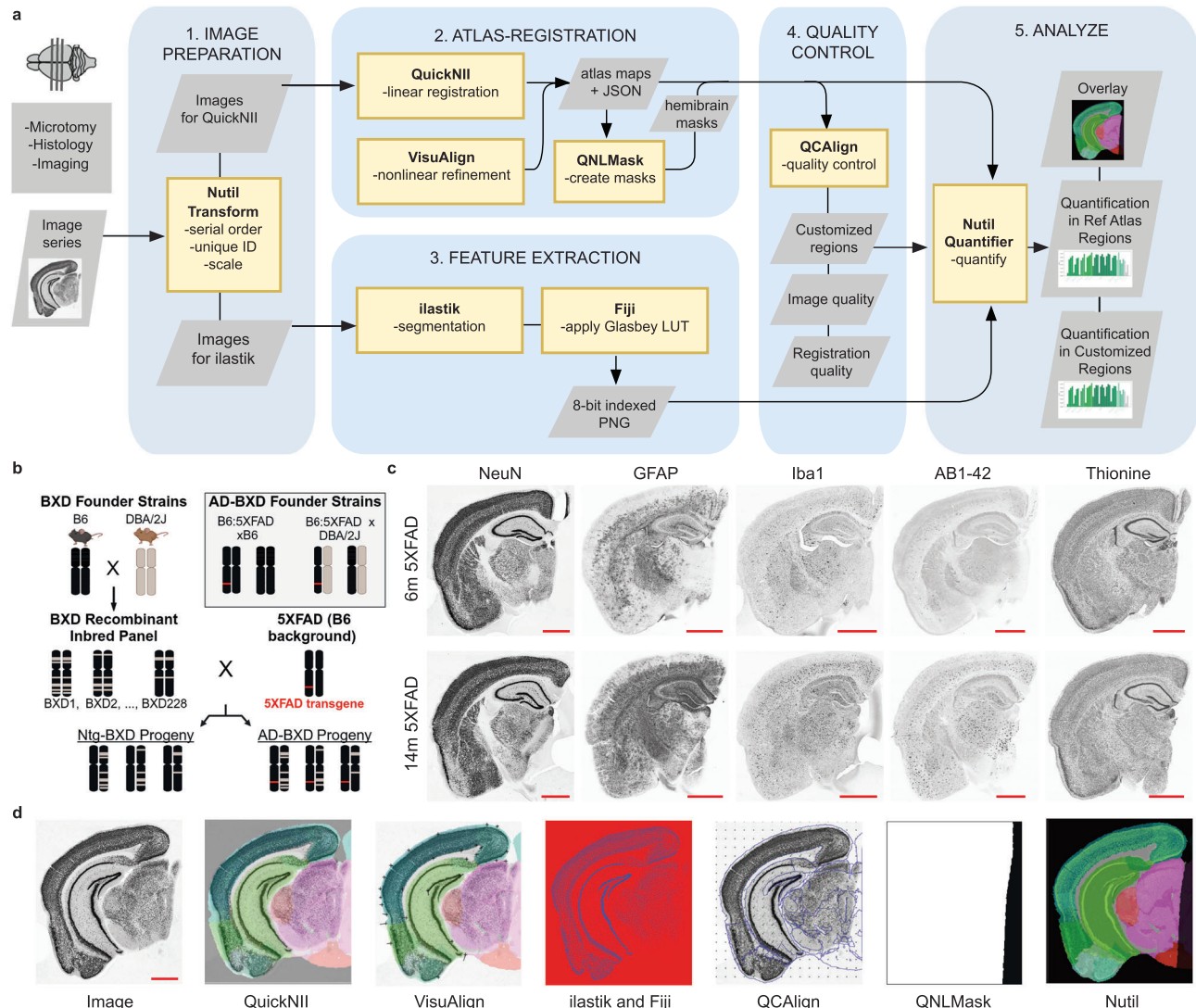

**Fig. 1 | Study design and QUINT workflow overview. a** Regional cell and amyloid-beta composition were quantified using the expanded QUINT workflow. (1) Raw images were processed to meet size requirements. (2) Brain sections were registered to the Allen Mouse Brain Atlas CCFv3 2015 in QuickNII and refined using VisuAlign. Hemibrain masks were created in QNLMask (3) Ilastik pixel classification was used to segment the images for the features-of-interest for each stain and converted to RGB format in FIJI. (4) Post-registration quality control assessment was performed using the QCAlign tool. (5) The output of the segmentation, registration, and mask creation steps were combined using Nutil to determine the percent stain-positive coverage per region area. **b** Immunohistochemistry was completed for an experimental cohort of 40 mice from the AD-BXD mouse model of AD (see Supplementary Table 1). Adapted from Neuner et al., 2019[9,90]. Created with BioRender.com. **c** Representative images of brain sections of 6 m and 14 m mice were sectioned and stained for thionine, NeuN, GFAP, Iba1, and AB1-42. The red scale bar on each image represents 1000 μm. **d** Representative images from each step in the QUINT workflow. B6 C57BL/6J, Ntg Nontransgenic.

ability of QUINT to achieve highly accurate registration to the regions selected for the study.

## Nonlinear adjustment increases regional registration accuracy and impacts cell and amyloid-beta load estimates

VisuAlign provides users with an accessible graphical user interface (GUI) where they can systematically make nonlinear adjustments to regional boundaries of the atlas-plates from the Allen Mouse Brain Atlas as set in QuickNII, to reflect the structural composition of the experimental section more accurately (Fig. 2a). This process involves identifying mismatches between the atlas-plate and the underlying experimental section and manually positioning and dragging anchor points on the atlas-plate to their correct position on the section. The importance of nonlinear adjustment following linear registration is demonstrated by comparing the QCAlign results following each atlas-registration step in the QUINT workflow (Fig. 2b). Subregions of the hippocampal formation are particularly vulnerable and require more extensive nonlinear adjustment due to their distinct shape, relatively small size, and distinctive cell layers (Fig. 2b inset). Mean regional accuracy scores of five brains were calculated and compared following atlas-registration performed using QuickNII alone (rated by 2 individuals) relative to the registration performed using QuickNII followed by adjustment with VisuAlign (rated by 6-10 individuals). The use of VisuAlign greatly improved atlas-registration to the brain sections (Fig. 3a, green vs navy). Regions with the greatest increases in accuracy scores included those not prioritized when initially aligning atlas-plates to the brain sections in QuickNII, thereby requiring more extensive nonlinear adjustment (e.g., regions comprising the mid- and hindbrain). Regional quantification of cellular and amyloid-beta load was also impacted by the increased registration accuracy achieved following nonlinear adjustment. Regions that required the most adjustment in VisuAlign also had the greatest difference in load values when comparing regional cellular and amyloid load output achieved using QuickNII alone versus registration completed in QuickNII and refined in VisuAlign (Fig. 3b, Supplementary Data 3). In conclusion, the capability to perform nonlinear adjustments to QuickNII atlas-registration

**Fig. 2 | VisuAlign and QCAlign were used to refine and verify the regional atlas-registration achieved in the QUINT workflow. a** VisuAlign GUI displaying one thionine section with nonlinear refinements applied to achieve an improved match of the atlas delineations over the section. CCFv3 regional borders are overlaid on the section with the position of the borders manipulated using anchor points. The lines indicate the start position of the points prior to nonlinear refinement, with the black markers denoting their final position after nonlinear refinement. (i). Inset displaying the atlas-registration achieved by linear registration using QuickNII. The dentate gyrus cell layers are incorrectly positioned over the section. (ii). Inset displaying the atlas-registration achieved using QuickNII and VisuAlign. The positioning of the dentate gyrus cell layers has been adjusted to match the cell layers in the section. **b** QCAlign GUI displaying one thionine section with a grid of systematic random sampling points overlaid. Grid points are marked up as registered accurately (+) or inaccurately (−) based on the region name, which is displayed in the GUI by hovering over a point (region name shown for point indicated with the arrow). (iii). Inset displaying the quality of the atlas-registration achieved by linear registration with QuickNII only (87% accurate for the inset) (iii). Inset displaying the quality of the atlas-registration achieved by registration with QuickNII and VisuAlign (with nonlinear refinement) (100% accurate for inset iv.).

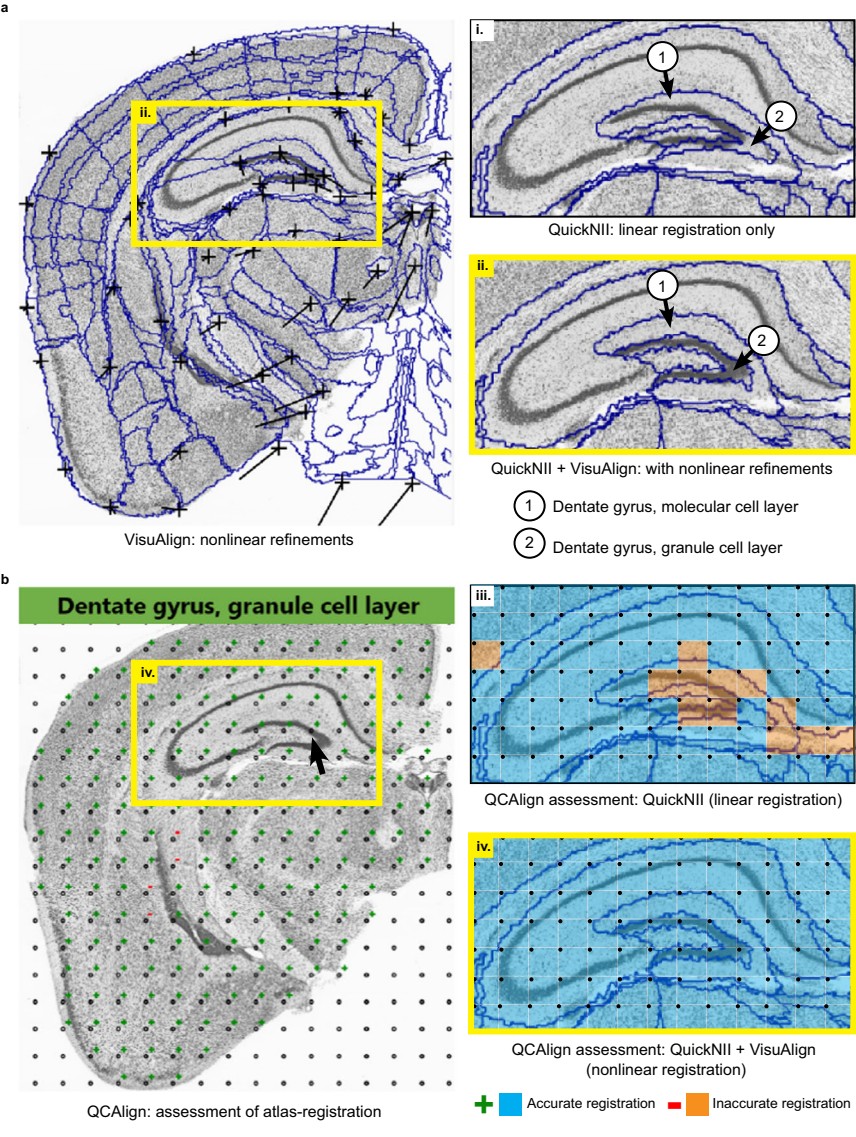

in VisuAlign is crucial because it improves regional registration accuracy leading to more reliable cell and amyloid-beta load estimates.

## AD-BXD mice exhibit widespread increases in glial and amyloid-beta accumulation from 6 m to 14 m

We compared differences in cell composition and amyloid-beta load between 5XFAD carriers of 6 m and 14 m to detect regional changes that occur with age and AD (Fig. 4, Supplementary Data 4, all $p$-values are FDR corrected). Overall, we observed only minor changes in NeuN load between 6 m and 14 m animals (Fig. 4a, i). The only regions that exhibited significant age-related ($p$-value < 0.05) decreases in NeuN load were the Ammon's horn ($p$-value = 0.0472) and dentate gyrus, polymorph layer ($p$-value = 0.00299). Slight, but significant ($p$-value < 0.05) age-related increases in NeuN load were observed in the posterior amygdalar nucleus ($p$-value = 0.0327) and striatum-like amygdalar nuclei ($p$-value = 0.0258). Increased glial proliferation and reactivity are also hallmark symptoms of AD progression with age. With this dataset, we confirmed that regional astrocyte (Fig. 4a, ii) and microglial cell load (Fig. 4a, iii) increased from 6 m to 14 m in AD-BXD animals. The caudoputamen exhibited the most significant increase in GFAP load ($p$ = 2.91E−10). The midbrain (motor-related) regions ($p$-value = 1.26E−08) and olfactory tubercle ($p$-value = 1.55E−08) exhibited the greatest

microglial load increase from 6 to 14 m. Aligned with previous reports in 5XFAD animals[37], amyloid-beta deposition was most prevalent within the subiculum at the 6 m time point (3.41 ± 0.227%, Fig. 4a, iv). In addition to the subiculum, amygdalar regions were highly susceptible to amyloid-beta deposition by adulthood (6 m). As an aggressive amyloidosis AD model, the AD-BXD animals exhibited a near-global increase in amyloid-beta deposition between 6 m and 14 m. Increased amyloid-beta deposition was strongly detected within the hippocampus and hippocampal-projected regions, including the cortex, thalamus, and amygdalar regions as previously noted (Fig. 4a, iv)[38].

## Individual AD-BXD strains exhibit variation in region neuronal load

While there were minimal significant regional differences in NeuN load between the age groups (only 2 regions exhibited a significant decrease in NeuN load (Fig. 4a, i)), we observed age-related strain-specific variation in NeuN load in hippocampal subregions (Fig. 4b, Supplementary Fig. 3). AD-BXD strains ranged from neurodegeneration to neuronal maintenance between 6 m and 14 m, modeling the heterogeneity observed in human AD[39]. No strain effect was detected in stain load among the 43 intermediate atlas regions quantified (uncorrected and FDR $p$-value > 0.05, 2-way ANOVA), but since sample sizes per strain were relatively

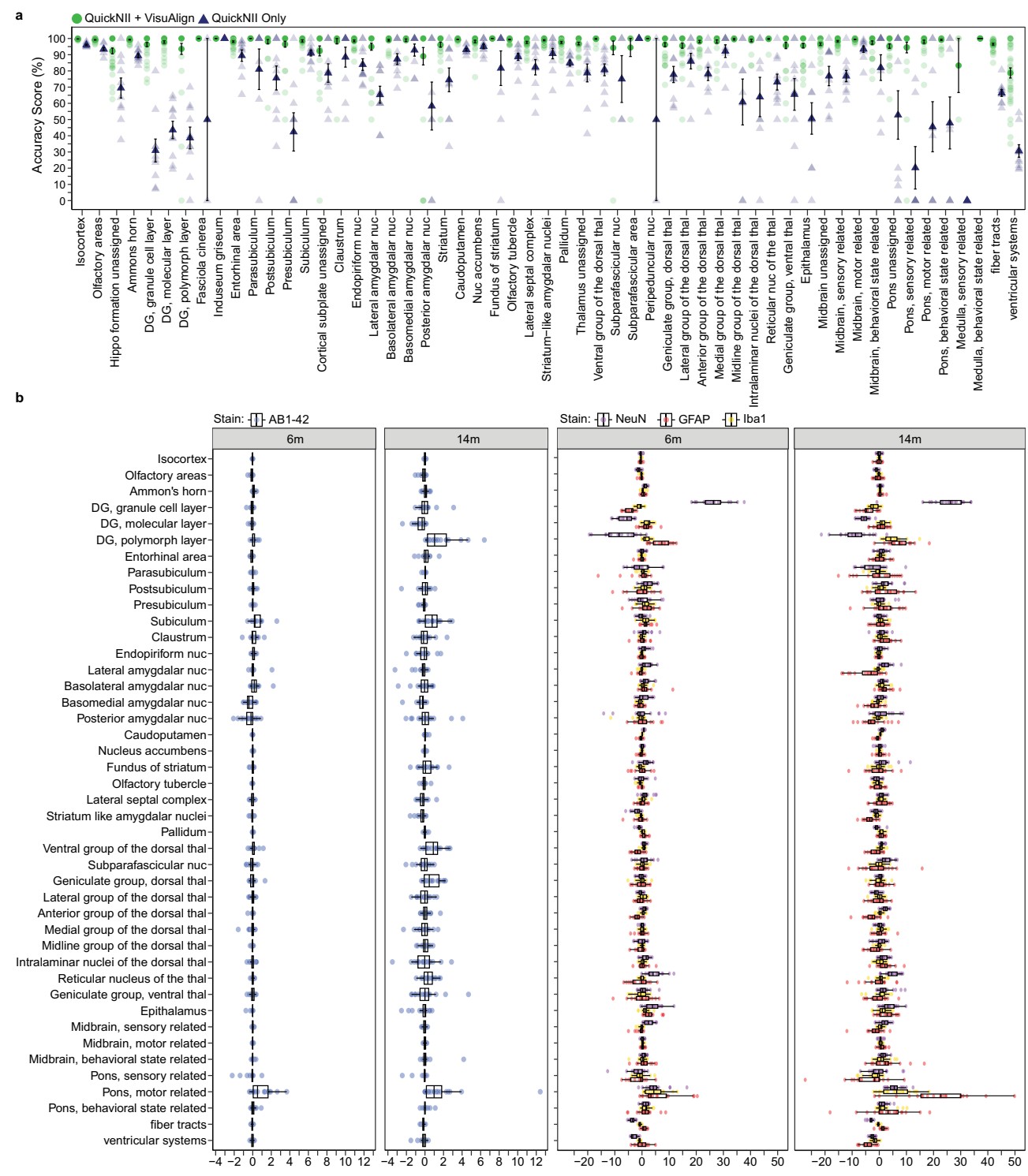

**Fig. 3 | QCAlign verification of regional atlas-registration at the selected intermediate hierarchy level. a** Mean accuracy scores per intermediate hierarchy region after QuickNII registration alone (navy triangles) or after QuickNII and VisuAlign registration (green circles). Two raters scored the same 5 randomly selected brains after QuickNII registration alone, max $n = 10$ assessments per region (Raters: $n = 2$ per brain). Up to 10 raters scored the same 5 randomly selected brains after QuickNII and VisuAlign registration, max $n = 36$ assessments per region (Raters: $n = 6$–10 per brain, see Supplementary Data 2 for the exact number of assessments measured per region). Dark shapes represent the mean score across raters per region for 5 brains +/− SEM, with the opaque shapes representing the individual assessments

contributing to each mean calculation. **b** The impact of VisuAlign refinement on regional stain load (%-stain-positive coverage/per region area) was measured by calculating the difference in load following Nutil quantification after each method (regional (QuickNII + VisuAlign output (%)) – regional (QuickNII output (%)) = regional load difference (%)). Boxplots of individual regional differences in load values are represented for all 5XFAD animals at 6 m and 14 m (6 m: $n = 17$ mice, 14 m: $n = 20$ mice). Dots represent mean regional load difference +/− SEM for all 5XFAD animals at 6 m and 14 m (6 m: $n = 17$ mice, 14 m: $n = 20$ mice). DG dentate gyrus, Nuc nucleus, Thal thalamus.

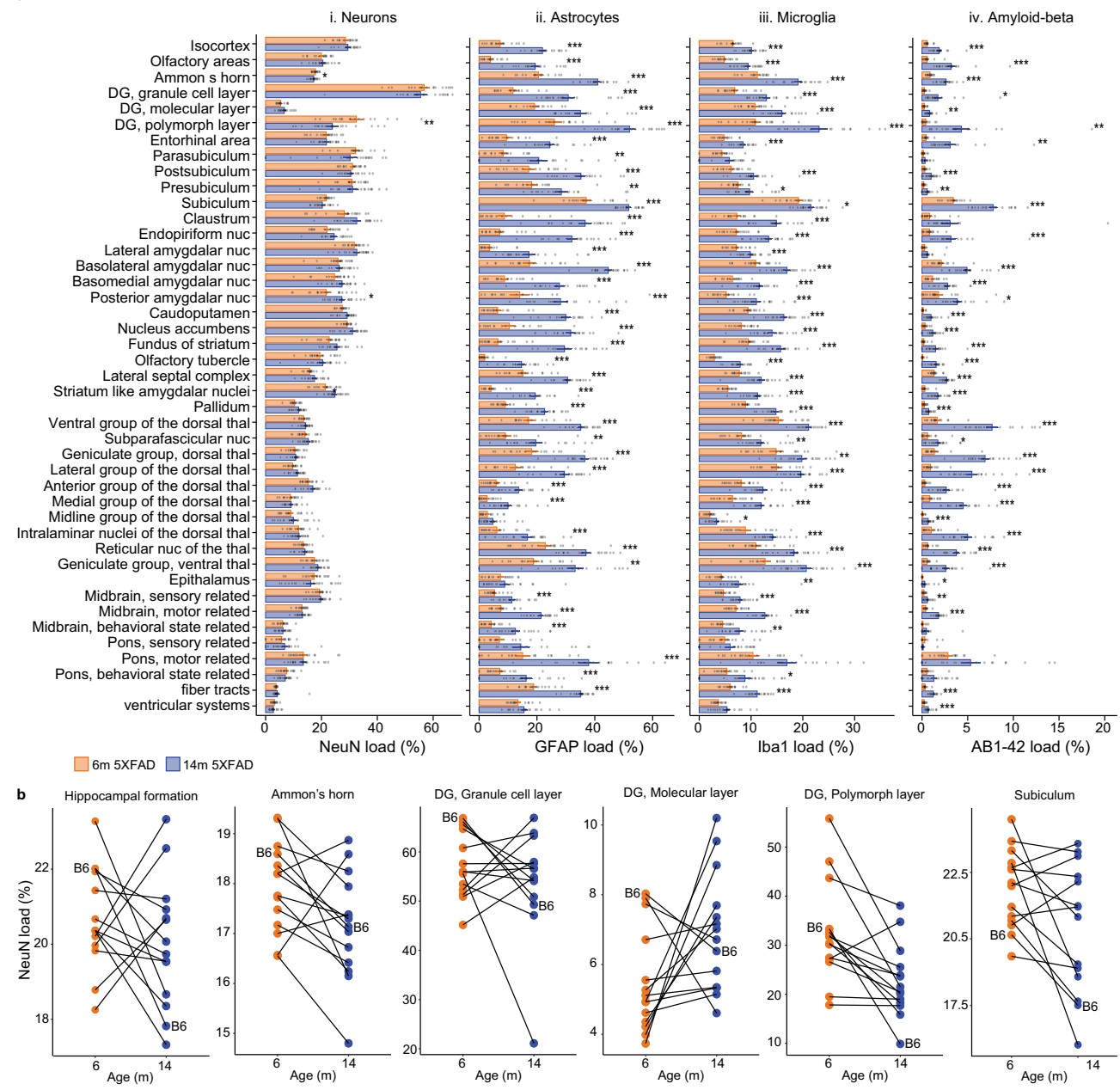

**Fig. 4 | Regional cell and amyloid-beta load varies from adulthood (6 m) to middle age (14 m) in AD mice. a** Regional cell and amyloid-beta load of the intermediate hierarchy regions of 5XFAD mice. i. Differences in NeuN load between the age groups were limited across the intermediate hierarchy regions. ii.-iv. GFAP, Iba1, and AB1-42 load increased with age across most intermediate hierarchy regions. Bars represent regional averages +/- SEM for 6 m and 14 m groups. Individual points represent regional load values for individual mice (5XFAD mice only, 6 m: $n = 17$ mice, 14 m: $n = 20$ mice). FDR corrected $p$-values represented. $P$-value: * <0.05, ** <0.01, *** <0.001. **b** Strain averages of NeuN load across the hippocampal formation and hippocampal intermediate hierarchy subregions. Points are mean load per strain. Lines connect strain matches across the two age groups: 6 m and 14 m. Only strains with an aged match counterpart are represented (5XFAD mice only, $n = 14$ strains per age group, $n = 1$-3 biological replicates per strain, 6 m: $n = 17$ mice ($n = 1$ mouse/strain (B6 x BXD100, BXD44, BXD51, BXD60, BXD61, BXD62, BXD65, BXD69, BXD75, BXD77, BXD87, and C57BL.6J strains), $n = 2$ mice/strain (B6 x BXD32), $n = 3$ mice/strain (B6 x DBA.2J)), 14 m: $n = 18$ mice ($n = 1$ mouse/ strain (B6 x BXD100, BXD44, BXD51, BXD60, BXD61, BXD62, BXD69, BXD75, BXD77, BXD87, and C57BL.6J), $n = 2$ mice/strain (B6 x BXD32 and BXD65), $n = 3$ mice/strain (B6 x DBA.2J))). The B6 founder strain is labeled for reference.

small in this analysis, a potential strain effect cannot be excluded, and will be evaluated in future analyses. Understanding that genetics strongly contribute to variation in symptom onset and susceptibility to AD in both humans[40,41] and AD-BXD mice[9,42,43], here we have highlighted the translatable potential to investigate the influence of genetic background on the presentation of neurodegeneration in animals carrying the 5XFAD transgene.

**Integration of paired IHC and bulk RNA sequencing data reveals cell load is a confounding factor in age-by-gene expression correlations among AD-BXDs**

Due to the inherent properties of bulk RNAseq, which measures tissue-averaged gene expression, the influence of cell composition is often overlooked in the interpretation of gene expression data and may conflate expression differences driven by other experimental factors

such as age and pathology accumulation[17,19,44,45]. Using the output from our QUINT analysis, we integrated hippocampal formation cell (NeuN, GFAP, Iba1) and pathology (AB1-42) load output with gene expression data measured via bulk RNAseq obtained from the contralateral hippocampal formation of the same 5XFAD mice at two ages (previously published[9,10,12]).

Hippocampal load per stain type was correlated with normalized read counts to identify age-dependent relationships between load and gene expression. The percentage of the 15,703 genes analyzed in the RNAseq dataset whose expression was significantly correlated (uncorrected $p$-value < 0.05) with load varied by stain type (NeuN: 16.35%, GFAP: 36.76%, Iba1: 34.78%, AB1-42: 31.86%) (Fig. 5a). Notably, stains that had

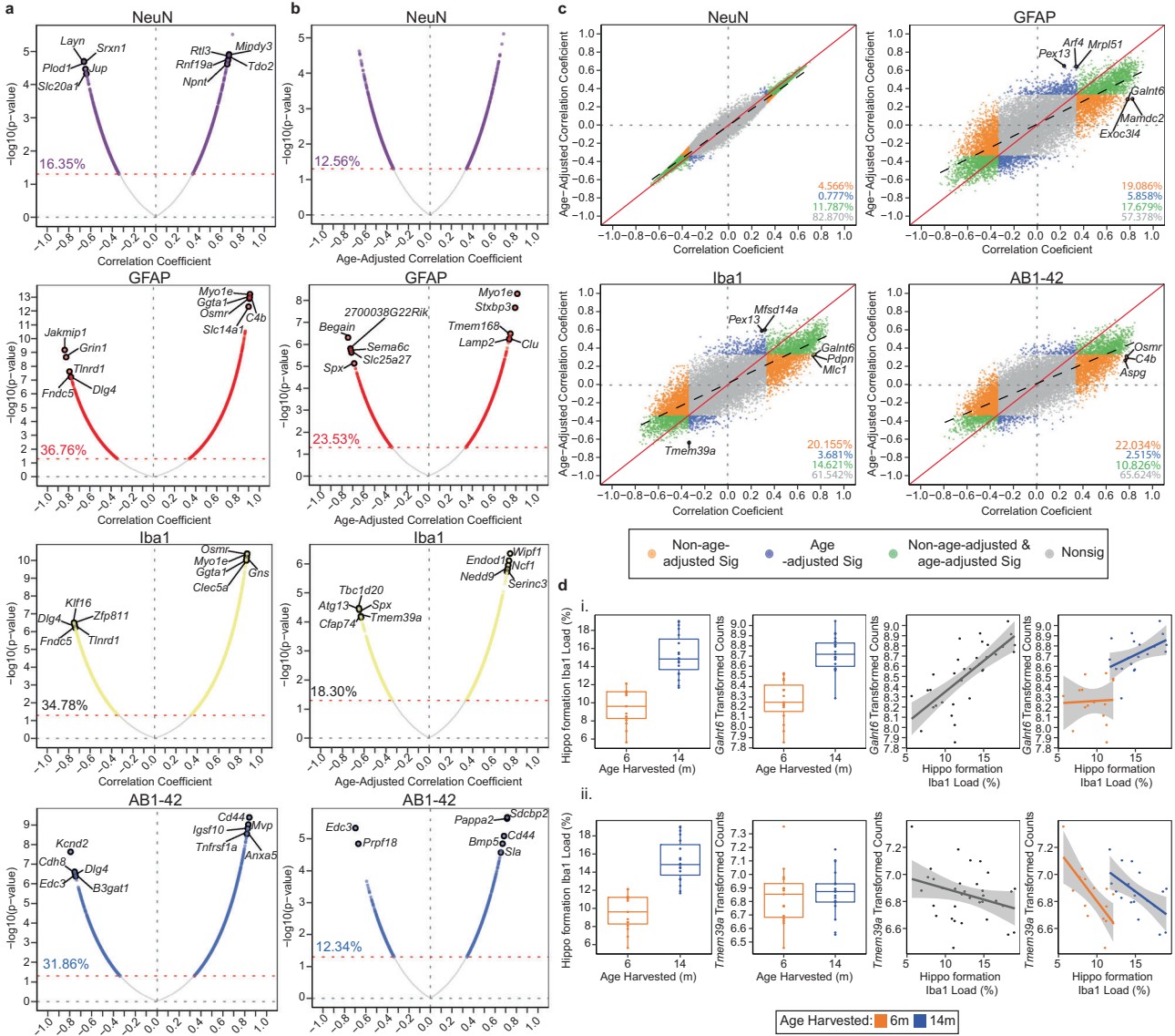

**Fig. 5 | Stain-specific load correlations with RNAseq gene expression to identify genes impacted by changes in load within the hippocampal formation. a** Gene expression by load Pearson R correlation coefficients and $p$-value relationships without age adjustment for each stain. Significantly correlated genes (uncorrected $p$-value < 0.05) are colored in each plot. The percentage of uncorrected significant genes is indicated within the plot. The top five positive and negative FDR significant (FDR $p$-value < 0.05) correlated genes are labeled. **b** Gene expression by load Pearson R correlation coefficients and $p$-value relationships after age adjustment for each stain. Significantly correlated genes (uncorrected $p$-value < 0.05) are colored according to stain. The percentage of uncorrected significant genes is indicated within the plot. The top five positive and negative FDR significant (FDR $p$-value < 0.05) correlated genes are labeled. **c** Comparison of Pearson R correlation coefficients without and with age adjustment per stain. Gene correlations that were exclusively significant (uncorrected-$p$-value < 0.05) without age adjustment are considered age-dependent (orange). Gene correlations that were exclusively significant (uncorrected-$p$-value < 0.05) with age adjustment are considered age-independent (blue). The specific influence of age and load cannot be disseminated in

gene correlations that were significant (uncorrected-$p$-value < 0.05) under both correlation conditions (green). All nonsignificant (uncorrected-$p$-value < 0.05) genes are labeled in gray. The percentage of significant genes per category is represented in the bottom right corner. The top 3 most significant genes per correlation method category are labeled per stain plot (FDR $p$-value < 0.05). **d** Individual relationship between gene expression and load with age for the top age-dependent and independently correlated genes with Iba1. i. *Galnt6* was exclusively significantly correlated with Iba1 without age adjustment. An increase in Iba1 load and *Galnt6* expression occurs between 6 m and 14 m. A positive relationship between Iba1 load and *Galnt6* expression exists across both age groups as well as within each age group. ii. *Tmem39a* was exclusively significantly correlated with Iba1 after age adjustment. An increase in Iba1 load but not in *Tmem39a* expression occurs between 6 m and 14 m. A weak relationship between Iba1 load and *Tmem39a* expression exists across both age groups, but separate age-specific correlations with load and gene expression exist. 5XFAD mice only, 6 m: $n$ = 17 mice, 14 m: $n$ = 20 mice.

the most significant gene correlates had the greatest age-related changes in load. Since our population is comprised of two ages and age was the primary driver of variation in load (Fig. 4), we next sought to identify genes that are related to load in an age-independent manner. We tested the role of age as a mediator of the relationship between stain load and gene expression using a multilevel correlation approach adjusting for the effect of age. Like the outcomes of the age-dependent correlation above (Fig. 5a), the percentage of genes significantly correlated with load after age adjustment (uncorrected $p$-value < 0.05) also varied by stain type (NeuN: 12.56%, GFAP: 23.53%, Iba1: 18.30%, AB1-42: 12.34%, Fig. 5b). The number of gene correlates (uncorrected $p$-value < 0.05) was reduced following age adjustment across all stains, with AB1-42 exhibiting the greatest reduction of significantly correlated genes (19.52% reduction, Fig. 5a, b). Next, we sought to identify genes that were exclusively significantly correlated with load either before or after age adjustment. By comparing both analyses (age-unadjusted, Fig. 5a and age-adjusted, Fig. 5b), we classified genes into (1) exclusively associated with variation in load in an age-dependent manner (non-age-adjusted output (orange in Fig. 5c)), (2) exclusively associated with load irrespective of age (age-adjusted output (blue in Fig. 5c)), or 3) associated with both load and age (non-age-adjusted and age-adjusted output (green in Fig. 5c)) (Supplementary Data 5). The majority of the significant correlations between gene expression and load were driven by age as indicated by the greater abundance of non-adjusted significant genes per stain (Fig. 5c). This age-driven relationship is illustrated by the correlation between Iba1 load and polypeptide N-acetylgalactosaminyltransferase 6 (*Galnt6*) expression, which was identified to be a top gene positively associated with variation in Iba1 load in an age-dependent manner (Fig. 5d, i). *Galnt6* has been found to have increased mRNA expression in the brains of AD patients and be related to amyloid-beta production[46,47]. Here, *Galnt6* exhibited increased expression with age that parallels the increase in Iba1 load observed from 6 m to 14 m (Fig. 5d, i, ii). On the contrary, only 0.78%-5.86% of the gene correlates per stain were exclusively significant only after age adjustment, indicating that these genes are likely associated with load in an age-independent manner (Fig. 5c). These significant age-independent genes exhibited a pattern of increased cell (GFAP and Iba1) and pathology (AB1-42) load but no difference in gene expression between 6 m and 14 m. This pattern is exemplified by the relationship between gene expression and load with age for transmembrane protein 39 A (*Tmem39a*), a topmost correlated gene with Iba1 load after age adjustment (Fig. 5d, ii). *Tmem39a* is a known contributor to pathways implicated in AD, including inflammation, dysregulated type I interferon responses, and other immune processes[48]. *Tmem39a* exhibited specific within-age-group associations between load and gene expression (Fig. 5d, ii). *Tmem39a* and other genes with significance following age adjustment may be driven by load differences between groups independent of the effect of age on load. We found that the relationship between gene expression and GFAP, Iba1, and AB1-42 load is consistent among the topmost significant correlated genes following FDR correction for each adjustment method (not age-adjusted or age-adjusted). Genes exclusively significant prior to age adjustment exhibit an age-related increase in gene expression that mirrors the age-related increase in hippocampal formation AB1-42, GFAP, and Iba1 load. Similarly, when evaluating the most significantly correlated genes with GFAP, AB1-42, or Iba1 load exclusively after age adjustment, we saw a consistent pattern of increased regional stain load with age without age-related increases in gene expression.

In conclusion, our analysis demonstrates that variations in cell and amyloid-beta load can significantly affect the interpretation of age-by-gene expression correlations. This highlights the importance of considering cell composition as a potentially confounding factor, particularly in studies involving age-related diseases like AD. By separating age-dependent and age-independent gene correlates, we could better distinguish between genes whose expression changes directly with age and AD pathogenesis versus those whose expression changes are driven by age-related alterations in cell populations. This distinction helps inform whether candidate gene expression (e.g., overexpress or knockdown gene expression) or cell/amyloid-beta composition (e.g., target the maintenance of a cell type's load)

should be targeted. This analysis also provides valuable insights into the complex interplay between aging, cell composition, and gene expression in AD.

## Mediation of age reveals differential overrepresentation of Reactome pathways

Next, using the correlation coefficients in Fig. 5a and b, gene set enrichment analysis (GSEA)[49] was performed via WebGestalt[50,51] to extract biological insights from genes of interest and ultimately identify pathways biased by individual differences in cell and amyloid-beta load (Fig. 6). The gene Ensembl IDs and associated correlation coefficients calculated via the age-dependent or age-independent multilevel correlations discussed above were input into WebGestalt. The output normalized enrichment scores adjusted for multiple test corrections (FDR) were evaluated to determine whether gene sets for biological pathways are enriched among the positive and/or negative multilevel correlations. As expected, immune pathways were highly positively enriched for GFAP, Iba1, and AB1-42 correlations. We also observed a negative relationship between the enrichment of neuronal pathways and GFAP, Iba1, and AB1-42, highlighting the potentially detrimental impact these cell types have on neuronal functioning in the context of AD. Fewer significantly enriched pathways were associated with NeuN load (age-adjusted and non-age-adjusted), consistent with the subtle changes in load between 6 m and 14 m 5XFADs. The most highly enriched pathways for each stain and correlation method were involved in chromatin organization, extracellular matrix organization, immune system, metabolism of RNA, and the neuronal system. In comparing enriched pathways for age-adjusted and non-age-adjusted correlations per stain, the greatest difference in enriched pathways was observed within the cell cycle category for Iba1, GFAP, and AB1-42 stain types. Enrichment of these pathways is consistent with the proliferation of these cell types and amyloid-beta and the potential increase in immunoreactive cell cycle proteins with age[52]. A total of 42 cell cycle pathways were represented across these stain types after age adjustment, while only 2 were present prior to adjustment. Moreover, many negatively enriched pathways including those in the gene expression (transcription) and metabolism of RNA parent pathways were observed almost exclusively within the non-age-adjusted category for GFAP, Iba1, and AB1-42. This pattern of enrichment suggests a more pronounced involvement of these pathways with AD-related deterioration with age than necessarily with increased glial and amyloid-beta composition[53,54]. Ultimately, by using these methods we have begun to disseminate the effects of cell and amyloid-beta composition in the hippocampal formation and their implication in biologically relevant pathways.

## Discussion

Here, we report brain-wide IHC output from 37 mice of the AD-BXD panel obtained using the QUINT workflow. The increased functionality in the QUINT workflow enhanced the atlas-registration and enabled quality control assessments, increasing the quality of regional quantification. We quantified age-related differences and characterized the influence of genetic diversity among AD-BXD strains on NeuN, GFAP, Iba1, and AB1-42 load across a validated list of subregions from the CCFv3 2015. Lastly, the importance of accounting for variation in tissue cell and amyloid-beta composition was emphasized by integrating hippocampal load output with RNAseq data.

Overall, we demonstrate the capacity of the QUINT workflow to effectively detect subtle differences in regional loads accurately across the whole brain, which is paramount in the context of high-throughput imaging studies that incorporate genetic diversity models of disease. VisuAlign provides the capability to make nonlinear adjustments to the linear atlas-registration achieved using QuickNII, thus correcting for distortions in the sections introduced during IHC section preparation as well as for structural differences among regions in diverse disease models and age groups[31-35]. The importance of applying nonlinear refinements was demonstrated by quantifying the regional differences in accuracy scores and stain load achieved with QUINT using QuickNII only, relative to registration using

**Fig. 6 | Gene Set Enrichment Analysis (GSEA) of gene correlations per method categorized by Reactome parent pathway.** Pearson R correlation coefficients from Fig. 5a and b were input into WebGestalt GSEA to obtain significantly enriched pathways associated with each stain and correlation method (normalized enrichment, non-age-adjusted and age-adjusted). The top three most significant pathways per stain and methods are labeled (FDR *p*-value < 0.05) (right).

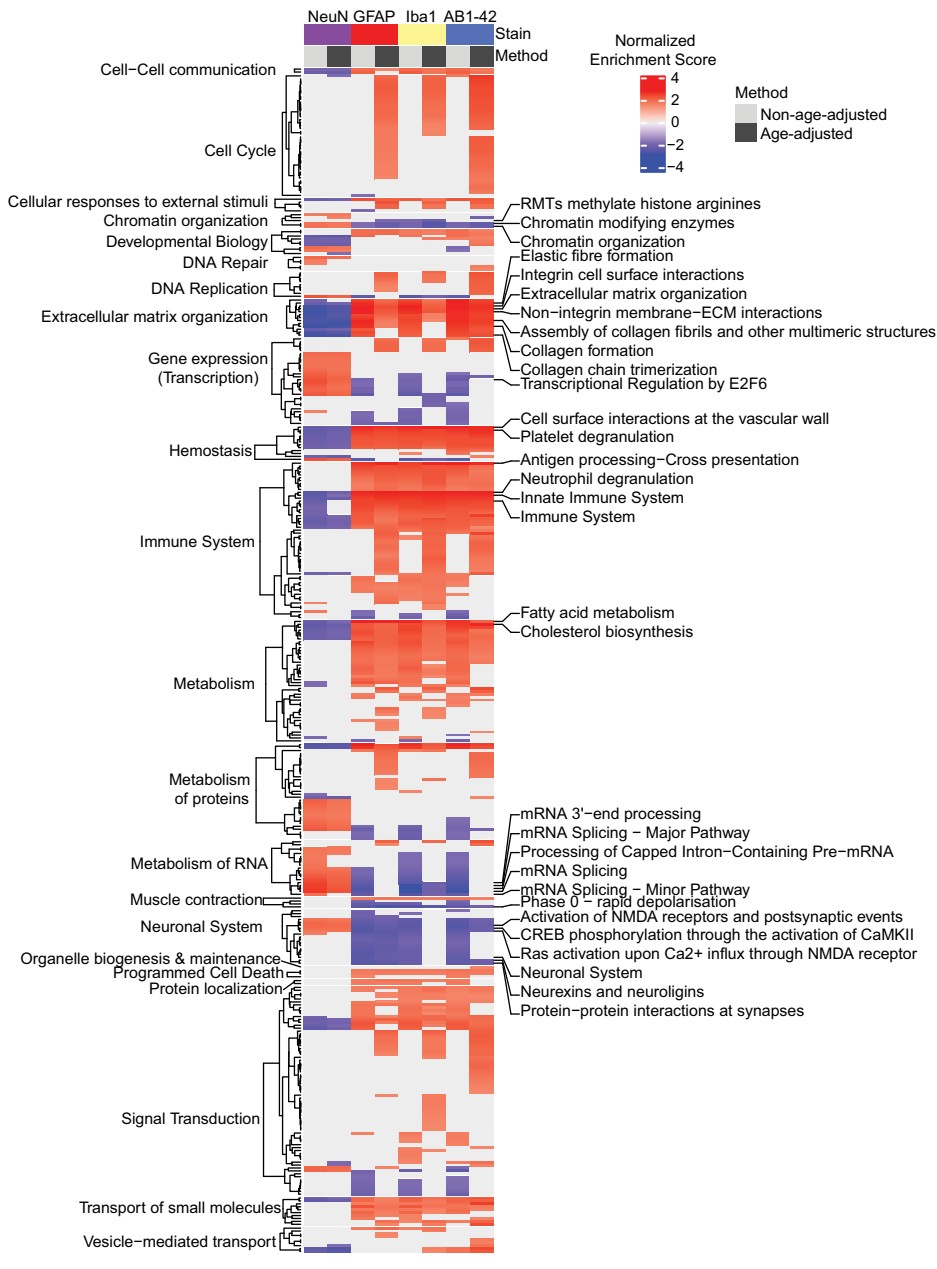

QuickNII and VisuAlign. The QCAlign tool enabled validation of the atlas-registration to the regions selected for the study, which was important since changes driven by genetic differences across strains were predicted to be subtle and region-specific. The high accuracy and low variability of reported QCAlign scores amongst raters and brains assessed heightened our confidence that the present cohort of brains was registered to a high standard. Another key functionality of QCAlign is its ability to produce customized hierarchies via parsing through and selecting reference regions to compile into related summary regions. This is an approach also used in other studies to compensate for the difficulty in registering regions that lack anatomical boundaries[55–57]. Defining a standardized regional hierarchy also promotes the labeling of consistent regions of interest among laboratories.

The QUINT workflow has numerous advantages over alternative methods. It promotes comprehensive regional analysis as defined by a standardized reference atlas, facilitating comparison, integration, and reproducibility of results across studies in compliance with the FAIR guiding principles[24,25]. While traditional IHC methods that rely on the manual delineation of regions and counting via stereology are more commonly used in

the field, they are inefficient for brain-wide exploration in studies with large numbers of animals[58–61]. As demonstrated here, the QUINT workflow supports large-scale comparative studies[62] and has the capacity to characterize transgenic models of diseases of varying strains, ages, and genotypes. The ability to share the intermediate results of the workflow provides transparency, which is important since atlas-registration and feature extraction are inherently subjective processes guided by user-based expertise. This subjective nature of the registrations is counteracted with the addition of QCAlign which provides a means to evaluate the atlas-registration. The main downside of QUINT is that it can be labor intensive, especially for damaged and distorted sections as nonlinear adjustments must be applied manually to match deviations from the atlas-plates established in QuickNII. Although this is time-consuming, we demonstrate that it is important since it considerably improves the quality of atlas-registration, as well as the quality of regional results. Efforts to further automate the atlas-registration step are underway using deep neural networks (DeepSlice)[27].

The QUINT workflow is a powerful approach for the high-throughput exploration that is needed to unravel the complexity of AD. Using this

approach, we validated the severity of neuroinflammation and amyloid-beta accumulation in the brains of aging 5XFAD animals[37,63–65] and expanded the scope of regions investigated in this diverse AD population[66]. Near global increases in AB1-42, GFAP, and Iba1 were reported as the AD-BXD mice aged from 6 m to 14m[37,65]. We also demonstrate that regions that exhibited neurodegeneration, like the Ammon's horn, were among those that exhibited the greatest increase in amyloid-beta and neuroinflammation. In contrast to the initial qualitative[37] and later quantitative[63–65] studies that identified neuron loss in cortical layer V and subiculum starting at 6 m in 5XFADs compared to controls, our study, due to the nature of the dataset, could only detect age-related neurodegeneration in 5XFADs. While these studies did not measure neurodegeneration in the hippocampus overall, more recent studies quantified a decrease in NeuN protein in the hippocampus by 8 m[67], 10 m[68], and 12 m[66] in 5XFAD mice compared to nontransgenic animals (Ntgs). The differences in age, sex, and genetic background as well as analytical differences amongst studies (e.g. regions compared) may explain the discrepancy in the detection of neurodegeneration in these studies and ours. Since we studied the effects of the 5XFAD transgene using the diverse AD-BXD panel, we were uniquely positioned to detect variation in NeuN load among AD-BXD strains. We demonstrate a trend that certain AD-BXD strains exhibit a decrease in NeuN load from 6 m to 14 m while other strains do not. We believe that this NeuN variation is not due to varying APP expression levels as we have previously measured transgene expression via quantification of human APP expression and endogenous mouse APP levels and found no significant differences among the panel of strains evaluated[9]. Instead, we hypothesize that genetic differences amongst the AD-BXD strains may influence how each strain copes with the effects of the 5XFAD transgene and aging[2,9,10,43]. Ultimately, AD-BXD strains can be classified into general AD subtypes[69] or stratified as resilient or susceptible to AD pathology[11]: with resilient strains potentially mitigating neuron loss in response to neuroinflammation and pathology accumulation or staving off severe pathology accumulation altogether. Moreover, the AD-BXD panel has proven to be a robust population for genetic mapping of behavioral traits[9–12,70,71], and ongoing experiments performing quantitative trait loci mapping aim to elucidate genetic factors responsible for variation in heritable regional cell and amyloid-beta load[72,73].

Further, we demonstrate how QUINT results can be integrated with other data types, including omics data. RNAseq is a common method of profiling gene expression changes between different disease stages. However, results from bulk tissue samples reflect aggregate gene expression across heterogeneous populations of cells[15], meaning that expression differences may reflect cell-composition differences across tissue samples and/or true transcriptional differences across groups. Determining whether gene hits from bulk RNAseq data are driven by changes in transcriptional regulation or relative proportions of cell types in the samples is crucial to establishing and properly validating gene candidates of resilience or susceptibility to AD[11,18,74]. Recent AD case/control snRNAseq datasets offer the opportunity to better resolve such cellular differences[75], but have restrictive technical and cost constraints that can limit the size of such datasets in terms of cells collected and individuals sampled[76]. These limitations as well as the variable performance of deconvolution methods can make it difficult to establish cell-type specific differences in gene expression among heterogenous AD populations. While traditional methods for determining cell-type composition, such as IHC or flow cytometry, rely on a limited set of molecular markers and lack scalability relative to the current rate of data generation, the use of the QUINT workflow can expedite this process.

To combat these limitations of RNAseq, we integrated cell composition and RNAseq gene expression data using mixed modeling correlations. We were able to identify candidate genes associated with cell composition, dependent and independent of the effect of age on AD, thereby creating a guideline for subsequent analyses to distinguish whether gene expression or overall cell or amyloid-beta composition should be targeted. The resulting proportion of genes correlated with load stresses the importance of considering cell composition when analyzing RNAseq data. We also unmasked a unique subset of genes that exhibited no age-related changes in gene expression yet were correlated with variation in load within the age groups examined. Many of the genes that were exclusively significantly correlated with hippocampal formation load following age adjustment were enriched for cell cycle and immune system pathways. This study serves as proof of concept that IHC data, quantified by the QUINT workflow, can be used as a proxy for cell-type composition in the analysis of RNAseq data, and demonstrates that changes in gene expression may be relative to variation in cell composition with age and AD. Due to the nature of this dataset, our analysis was a partial mediation that begins to disentangle the effect of load, gene expression, and age with AD. Further unraveling this relationship, as well as the impact of diverse genetic backgrounds on the effects of the 5XFAD transgene and the identification of potential resilience mechanisms will require additional analyses including nontransgenic animals, males, and increased biological replicates.

A limitation of our study is the lack of consideration of tau pathology. Initially, the 5XFAD transgene was not thought to induce tau tangle pathology as no human tau transgene is expressed in this mouse line[37]; therefore, the 5XFAD model was prominently used as a model to investigate amyloid-associated neurodegeneration and neuron loss[64]. More recent investigations have reported the presence of varying pathogenic phospho-tau proteins at different tau residue sites in the brains of 5XFAD animals[67,77–81]. We have verified that AD-BXD animals exhibit strong amyloid and neuroinflammatory responses with age and AD, display a high level of concordance with both familial and sporadic forms of human AD at the molecular and behavioral level[9], and that female AD-BXD mice exhibit high translational alignment and conserved cell-type-specific signatures of resilience to AD with human AD cohorts[11], but technical limitations have impeded our ability to explore the presence of tau epitopes across the AD-BXD panel. These consistent findings characterizing tau in 5XFAD brains warrant the future investigation of phosphorylated tau in AD-BXD strains. Like the variation in NeuN we described here, we predict that strain-specific variation in tau may also be present in this panel.

Our study demonstrates that the QUINT workflow, with the addition of VisuAlign and QCAlign, proved to be a highly effective method for registering and quantifying cell and amyloid-beta deposition changes. Achieving high confidence regional output of AD-relevant cell types and amyloid-beta facilitated the exploration of genotype, age, and cell composition relationships. We provide the most detailed regional IHC characterization of 5XFAD mice known to date revealing age-related increases in amyloid-beta and glia as well as strain-specific variation in NeuN load. In response to characterizing the effects of age and genetic diversity in AD on cell composition and gene expression, we suggest that bulk-RNAseq data needs to be integrated with corresponding cell load values to generate robust and reproducible results that can be used to prioritize gene hits for future exploration. Using this method, we were able to reveal the enrichment of cell cycle and immune pathways in association with astrocytes, microglia, and beta-amyloid in an age-independent manner as the disease progressed from 6 m to 14 m. By achieving cell and amyloid-beta quantification in our AD-BXD population, we provide a framework for investigators to characterize diverse disease models and integrate their data with a range of behavior and/or omics data.

## Methods
### Bioethics
All mouse experiments occurred at the University of Tennessee Health Science Center and were carried out in accordance with the principles of the Basel Declaration and standards of the Association for the Assessment and Accreditation of Laboratory Animal Care (AAALAC), and the recommendations of the National Institutes of Health Guide for the Care and Use of Laboratory Animals. The protocol was approved by the Institutional Animal Care and Use Committee (IACUC) at the University of Tennessee Health Science Center. We have complied with all relevant ethical regulations for animal use.

## Animals

All data used in this study are from the AD-BXD panel including the founder strains[9] (Fig. 1b). This panel consists of female B6 mice hemizygous for the 5XFAD transgene (B6.Cg-Tg (APPSweF1LonPSEN1*M146L*L286V) 6799Vas/Mmjax, Stock No. #24848-JAX) mated to males from the BXD genetic reference panel to produce isogenic F1 AD-BXD strains carrying the 5XFAD transgene and their nontransgenic (Ntg)-BXD littermate "normal aging" controls. Male and female AD-BXD mice were group-housed as a mix of 5XFAD and Ntg same-sex littermates (2-5 per cage) and maintained on a 12-hour light-dark cycle with *ad libitum* access to food and water. All mice were genotyped for the 5XFAD transgene through a combination of in-house genotyping by The Jackson Laboratory Transgenic Genotyping Services (strain #34848-JAX) or by Transnetyx (TN, USA). This study included a total of 40 mice (2 males and 38 females) at 6 months (6 m; n = 20 mice) or 14 months (m) (14 m; n = 20 mice). These included 29 mice from 14 AD-BXD strains (n = 1–3 mice per strain per age group); 8 mice from founder strains C57Bl/6 J (B6)x B6:5XFAD (n = 2 mice), and F1 B6:5XFADxDBA/2J (B6/D2) 5XFAD (n = 6 mice); and 3 Ntg-BXD mice (all 6 m). An overview of all the animals in the study is given in Supplementary Table 1.

## Tissue Collection and Shipment

Mice were deeply anesthetized using isoflurane before decapitation and rapid removal of the brain at 6 m or 14 m. The hypothalamus was dissected out and the brain was bisected down the sagittal midline. One half of the brain was immediately further dissected and snap-frozen to be used for RNAseq. The other hemisphere was placed in 4% paraformaldehyde and kept at 4°C to be used for IHC[9,10,12]. To minimize technical variation in IHC, hemibrains were sent overnight to Neuroscience Associates (NSA) (Knoxville, TN), where the cerebellum was removed and hemibrains were embedded, processed, and stained simultaneously in blocks of 40 hemibrains.

## Neurohistology embedding and sectioning

Hemibrains received at NSA were treated overnight with 20% glycerol and 2% dimethylsulfoxide to prevent freeze artifacts. The specimens were embedded in a gelatin matrix using MultiBrain®/ MultiCord® Technology (NSA, Knoxville, TN). The blocks were rapidly frozen after curing by immersion in 2-Methylbutane chilled with crushed dry ice and mounted on a freezing stage of an AO 860 sliding microtome. The MultiBrain®/ MultiCord® blocks were sectioned coronally at 40 μm. All sections were cut through the entire length of the specimen and collected sequentially into a series of 24 containers. All containers contained Antigen Preserve solution (50% PBS pH7.0, 50% Ethylene Glycol, 1% Polyvinyl Pyrrolidone); no sections were discarded.

## IHC staining

Free-floating sections were stained for AB1-42 (amyloid-beta), glial fibrillary acidic protein (GFAP, astrocytes), and ionized calcium-binding adapter protein 1 (Iba1, microglia) on every 24th section spaced at 960μm, yielding approximately 9 sections per hemibrain. Staining for NeuN (neurons) and thionine (Nissl, cell bodies) was performed on every 12th section spaced at 480μm, yielding approximately 19 sections per hemibrain. For AB1-42, GFAP, Iba1, and NeuN, all incubation solutions from the blocking serum onward used Tris-buffered saline (TBS) with Triton X-100 as the vehicle; all rinses were with TBS. After a hydrogen peroxide treatment and blocking serum, sections were immunostained with primary antibodies listed in Supplementary Table 2 overnight at room temperature. Vehicle solutions contained Triton X-100 for permeabilization. Following rinses, sections were incubated with biotinylated secondary antibody, followed by ABC solution (avidin-biotin-HRP complex; VECTASTAIN® Elite ABC, Vector, Burlingame, CA). The sections were rinsed and then treated with diaminobenzidine tetrahydrochloride (DAB) and hydrogen peroxide to create a visible reaction product. Following further rinses, sections were mounted on gelatin-coated glass slides and air-dried. The slides were dehydrated in alcohol, cleared in xylene, and cover-slipped. For thionine-Nissl staining sections were mounted on gelatin-coated glass slides, air dried, and carried through the following sequence: 95% ethanol, 95% ethanol/Formaldehyde; 95% ethanol, Chloroform/Ether/absolute ethanol (8:1:1), 95% ethanol; 10% HCl/ethanol, 95% ethanol, 70% ethanol, deionized water, thionine (0.05% thionine/acetate buffer, pH 4.5) (Fisher, T40925), deionized water, 70% ethanol, 95% ethanol, Acetic Acid/ethanol, 95% ethanol, 100% ethanol, 100% ethanol, 1:1 100% ethanol/xylene, xylene (x2), coverslip.

## Imaging

NSA performed scanning of each slide at 20x using a Huron Digital Pathology TissueScope LE120 (0.4 μm/pixel). Brain image series were compiled by reconstructing the IHC sections as sectioned and indicated by NSA.

Further information and requests for resources and reagents should be directed to the Corresponding author.

## QUINT workflow development

The QUINT workflow enables brain-wide quantification of histological data relative to a reference atlas such as the CCFv3[36]. In the workflow (Fig. 1a), the QuickNII software[28] is used to spatially register atlas-plates from a 3D digital atlas to serial section images, ilastik[29] is used to extract features from the images (by segmentation), and Nutil[30] is used to quantify features per atlas-region. To meet the needs of the current project, two additional tools, VisuAlign (RRID: SCR_017978) and QCAlign (RRID: SCR_023088) were developed and integrated into the workflow. VisuAlign is used to apply in-plane nonlinear adjustments to the atlas-plates established in QuickNII to achieve the best fit over the sections. Nonlinear adjustments are made by identifying mismatches between the atlas-plate and the underlying section in the VisuAlign GUI, manually positioning anchor points on the atlas-plate, and dragging the points to their correct position on the section. VisuAlign then uses these anchor points to create a continuous, nonlinear deformation field covering the entire section image. QCAlign is used to 1) detect sections or regions not suited for QUINT analysis (i.e., due to damage), 2) assess the quality of the atlas-registration to the sections, and 3) explore atlas-hierarchy levels. All QCAlign assessments are performed by systematic random sampling. The second assessment is based on anatomical expertise by evaluating how well delineations supplied by the atlas match up with boundaries revealed by labeling. Since validation of the atlas-registration is only possible for regions that have boundaries visible in the sections, and reference atlases are structured in systematic hierarchies that group related regions[36], functionality is implemented for adjusting the hierarchy to a customized level that supports this validation (i.e. a level where the atlas delineations approximately match the boundaries that are visible in the sections).

## Image pre-processing

To prepare the images for segmentation with ilastik, they were inspected, cropped, and downscaled using different scaling factors for the different stains (AB1-42: 0.20, GFAP: 0.40, Iba1: 0.40, NeuN: 0.40, thionine: 0.35). Scaling factors were determined by gradually increasing the scaling factor and manually determining the level at which the images were maximally reduced without visually losing information and inducing blur. For the atlas-registration, images were downsampled to fulfill the image size requirements of QuickNII (scaling factor: 0.50) (detailed online under "Imaging preprocessing requirements")[82].

## Image Registration to the CCFv3 with QuickNII and VisuAlign

For each brain, QuickNII (RRID: SCR_016854, QuickNII-ABAMouse-v3-2015 version 2.2) was used to perform linear registration of serial section images (combined irrespective of stain) to the CCFv3 2015, followed by nonlinear refinement with VisuAlign (RRID: SCR_017978, version 0.8). For each image series, the thionine-stained sections were registered first since they provided the greatest visualization of region boundaries. Subsequently, all remaining sections were registered serially. Thionine staining was included to promote accurate atlas-registration and region detection but was not included in downstream analyses in this study. Two independent raters

verified the registration performed with QuickNII and the refinements made with VisuAlign.

## Cell Segmentation with ilastik

The ilastik software (RRID: SCR_015246) enables feature extraction by segmentation using supervised machine learning. For each stain, ten training images were loaded into the Pixel Classification workflow (v.1.3.3), two classes were created ("label" and "background"), and annotations of each class were applied in all the training images until the segmentation was deemed satisfactory and confirmed by two independent raters. The trained classifiers were applied to all the images of that stain using ilastik's batch-processing function. Segmentations were exported and colorized using the Glasbey Lookup Table in FIJI ()[83].

## Evaluation of section image quality with QCAlign

QCAlign (RRID: SCR_023088, version 0.7) was used to assess brain section integrity for each image series (all 40 brains were assessed) using a 5-voxel grid spacing. This involved marking up points that overlapped areas of damage (tissue tears, folds, artifacts, and errors in image acquisition) for all sections. Marker counts were exported and used to calculate the percentage of damage per section by dividing the number of damage markers by the total number of markers overlapping the section (damage = # damage markers per section / # of total markers per section). Section images with more than 30% damage were deemed unsuited for QUINT analysis (Supplementary Table 3). Nutil results per brain were re-calculated in R-Studio following the removal of results from the damaged sections.

## Creation of a customized atlas-hierarchy with QCAlign

Brain reference atlases such as the CCFv3 are organized in systematic hierarchies that group related regions[36]. A customized hierarchy level was created with QCAlign to use for quality assessment of the atlas-registration, and to define regions to be quantified with Nutil (hereafter referred to as the "intermediate hierarchy"). To create this intermediate hierarchy, the atlas delineations supplied by VisuAlign were overlaid on the thionine-stained sections at the finest level of atlas-granularity (full expansion of the CCFv3). A grid of points with a 15-voxel grid spacing was applied to the images, with the registration accuracy of each point marked up based on anatomical expertise ("accurate", "inaccurate" or "uncertain"). If a region received many "uncertain" markers due to obscure region boundaries, the hierarchy level was adjusted one level up; the process was repeated until the position of most of the markers could be verified (either "accurate" or "inaccurate"). The customized hierarchy was exported and used in the Nutil software to define the regions for quantification (Supplementary Data 1).

## Quality control assessment of atlas-registration to the section images using QCAlign

To determine the quality of the atlas-registration to each region in the intermediate hierarchy, ten raters (researchers in neuroscience and neuroanatomy) across two institutions were recruited to perform quality assessment using the QCAlign software. Assessments were performed on the atlas-registration achieved using QuickNII only (2 raters) and using both QuickNII and VisuAlign (10 raters). All assessments were performed on the thionine-stained sections from five brains (selected at random) at the established intermediate hierarchy level. To perform the assessment, markers with a 15-voxel grid spacing were overlaid on the sections and the position of each marker was assigned as either "accurate", "inaccurate" or "uncertain" using anatomical expertise. This was determined by inspecting the position of the marker with respect to landmarks, in comparison to the name of the region which was revealed by hovering over each marker (the atlas delineations were not visible during the assessment).

The QCAlign results were exported with marker counts indicated per region, per section, and per brain. Regional accuracy, inaccuracy, and uncertainty scores were calculated per rater per brain, and per brain overall in R-Studio (scripts shared on the BRAINSPACE GitHub repository[84]). Uncertainty scores were calculated by dividing the number of uncertain

markers by the total number of markers in the region, reflecting the region percentage for which the registration could not be verified (Uncertainty Score = (# uncertain markers)/(# accurate markers + # inaccurate markers + # uncertain markers). Since it was not possible to verify the registration of all the points (due to a lack of landmarks or limited expertise), the calculation of accuracy and inaccuracy scores correspond to the parts of regions for which the registration could be verified. Regional accuracy scores were calculated by dividing the number of accurate markers by the total number of accurate and inaccurate markers within the region (Accuracy Score = # accurate markers/ (# accurate markers + # inaccurate markers)). Mean scores were calculated by dividing the summed score of all assessments by the total number of assessments. For each region, the number of assessments contributing to the mean calculation depended on the number of raters and number of brains assessed, as well as how often accurate or inaccurate markers could be assigned by the raters (depending on the presence of grid markers in the region, tissue quality, and/or anatomical expertise). In some cases, regions were marked entirely as uncertain, excluding the assessment from the mean calculation. For the registration achieved with QuickNII only, 10 assessments were averaged across all raters and brains. For the registration achieved with QuickNII and VisuAlign, a maximum of 36 assessments were averaged across all raters and brains. See Supplementary Data 2 for the exact number of assessments measured per region.

## Regional quantification of stain load with Nutil

Nutil (RRID: SCR_017183) enables regional quantification of labeled features by applying the *Quantifier* feature to combine the output from the atlas-registration (QuickNII and VisuAlign) and feature extraction (ilastik) steps. Nutil (v0.7.0) was used to quantify the percentage of IHC-stained area per region area (hereafter referred to as "load") in the intermediate hierarchy regions per brain series. Since hemibrain sections were analyzed in the study, customized masks were created and used to exclude the atlas regions located in the missing hemibrain from the quantification. The hemibrain masks were created with the QNLMask software shared with the VisuAlign download. Nutil analysis was performed separately for each stain, with the quantification of neurons (NeuN), microglia (Iba1), astrocytes (GFAP), all nuclei (thionine), and amyloid-beta 1-42 (AB1-42) achieved according to the parameters defined in the NUT file shared in the BRAINSPACE GitHub repository[84]. The regional load values obtained from the Nutil reports were used in downstream analysis. Regional loads were quantified for QuickNII registration alone and following refinement with VisuAlign.

## Sample and region exclusion from post-analyses

One female 6 m mouse of AD-BXD strain 44 was removed from the downstream analysis because the majority of the sections were severely damaged prohibiting successful atlas-registration. While the 77 regions in the intermediate hierarchy are included in the Nutil reports[85], some of the regions did not give results since they were not present in the sections, or corresponded to a parent structure with results provided at a finer level of atlas-granularity. Regions with no biological results were disregarded from all analyses. In the present study, 55 of the regions were included in the QCAlign assessment of the atlas-registration across 5 brains; 43 of these regions were included in the assessment of cell and pathology load across 37 5XFAD brains. Region-specific exclusion criteria are reported in Supplementary Data 6.

## Bulk RNA sequencing

The RNAseq data used in the current study was previously published[9,10,12] and the dataset series (GSE) are accessible via the National Center for Biotechnology Information Gene Expression Omnibus (GEO) (GEO:GSE101144, GEO:GSE119215, GEO:GSE119408). The published data reported on the results of bulk RNA sequencing completed with snap frozen hippocampi from AD-BXD strains and Ntg-BXD littermate controls at 6 m and 14 m Neuner et al.[9] GEO accession number GSE101144 and (Neuner et al., 2019)[12] GEO accession number GSE119215: AD-BXDs: [6 m, $n = 71$ mice (47 females/24 males) and 14 m, $n = 86$ mice (45 females/41 males)], and Ntg-BXD counterparts: [6 m, $n = 31$ mice(17 females/14 males) and

14 m, $n$ = 33 mice (17 females/16 males)]), and (Heuer et al., 2020)[10] GEO accession number GSE119408: Ntg-BXD counterparts [6 m, $n$ = 27 mice (22 females/5 males) and 14 m, $n$ = 44 mice (28 females/16 males)])[9,10,12].

In brief, RNA was isolated using the RNeasy mini kit (QIAGEN) and treated with DNase to remove contaminating DNA. A BioAnalyzer (Agilent Technologies) was used to confirm RNA quality. All samples had RNA Integrity Numbers (RIN values) > 8.0. Sequencing libraries were prepared using Truseq Stranded mRNA Sample Preparation Kit (Illumina Inc.) and sequenced by 75 base pair paired-end sequencing on a HiSeq2500 (Illumina Inc). The GBRS/EMASE pipeline[86] was used to align reads to a diploid BXD transcriptome. An expectation maximization algorithm was used to align reads to the correct B or D allele.

Only 5XFAD samples with paired IHC and RNAseq data were selected ($n$ = 34 mice); therefore, all animals in this analysis had one hemisphere fixed for IHC and the contralateral hippocampus dissected for bulk RNAseq (Supplementary Table 1). Expected read counts (ERCs) were filtered to include genes with >10 ERCs in more than 50% of the samples from 5XFAD mice, resulting in 15,703 of 47,645 genes that passed filtering. After the exclusion of genes with low read counts, datasets were batch-corrected using the R/Combat-Seq package, then normalized and transformed using the default pipeline of R/DESeq2[87].

## Statistics and Reproducibility

For each stain, the load values of 43 regions were used for comparative analysis across 5XFAD brains at 6 m ($n$ = 17 mice) and 14 m ($n$ = 20 mice). Data are expressed as means ± standard error of the mean (SEM) or as otherwise indicated in graphs. Statistical analysis of data was performed using R version 4.0.0 (2020-04-24) -- "Arbor Day". Wilcoxon two-way assessment (strain and age factors) was implemented to determine if there were significant differences in the stain load as registered using QuickNII alone vs registered using QuickNII and VisuAlign. Analysis of variance (ANOVA) (age and strain factors) was used to determine whether there were significant differences in regional stain load between the 6 m and 14 m groups. Multilevel Pearson correlations with and without age corrections were used to evaluate the relationship between hippocampal stain load and gene expression. Multiple testing corrections for each test were performed using false discovery rate (FDR) correction. The criterion for measures to be considered uncorrected significant was $p$-value < 0.05 and significant after correction was FDR $p$-value < 0.05. Data normality was assessed using the Shapiro-Wilks method in R.

The relationship between gene expression and stain load (AB1-42, NeuN, GFAP, and Iba1) from the hippocampal formation (region "Hippo" in Supplementary Data 1) was assessed using Pearson's correlation from linear mixed models[88], which allowed the effect of age on the association between gene expression and load to be accounted for by including age as a random effect. $P$-values per stain and gene correlation were corrected for multiple comparisons via FDR correction and considered significant if the FDR $p$-value < 0.05. Genes that were exclusively significantly correlated (uncorrected $p$-value < 0.05) prior to age adjustment were deemed to be age-dependent correlates. Genes that were exclusively significantly correlated (uncorrected $p$-value < 0.05) following age adjustment were deemed to be age-independent correlates. Gene Set Enrichment Analysis (GSEA) was queried against Reactome pathways in WebGestalt[50,51] using the output correlation coefficients per gene and stain for each multi-level correlation method (age-adjusted and non-age-adjusted). Advanced GSEA parameters used included: Minimum number of IDs in the category: 20, Maximum number of IDs in the category: 2000, Significance Level: FDR < 0.05, and Number of permutations: 1000. Lastly, individual ERCs and hippocampal load data were incorporated into a DESeq model, and the design was run on the intercept (~1). Transformed normalized counts for boxplots in Fig. 5 were obtained using the DESeqDataSetFromMatrix() and counts() functions. Scripts used for RNA-seq normalization and modeling, IHC and RNAseq correlations, and visualization can be accessed on the BRAINSPACE GitHub repository[84].

The sample size per data point is indicated in each figure legend. QCAlign output measurements are defined as the number of assessments per region. Assessments per region are aggregated across multiple raters and distinct brains (see Supplementary Data 2 for the exact number of assessments measured per region). Age groups are defined in terms of individual mice (i.e., 6 m 5XFADs, $n$ = 17 mice). Biological replicates per strain are defined as multiple genetically identical mice of the same AD-BXD strain background. In our study, we had a range of $n$ = 1-3 biological replicates per strain (see Supplementary Table 1).

## Inclusion & Ethics

The authors are committed to making materials, data, code, and associated protocols promptly available to readers without undue qualifications.

## Sharing of QUINT tools and disclaimer

All the software in the QUINT workflow is open-source and shared on GitHub and nitrc.org under MIT license for QuickNII and VisuAlign; GNU General Public License (GPL) v3.0 for Nutil; and GPL v2 / GPL v3 for ilastik. To validate the QUINT workflow for the present study, Nutil v0.7.0 was used to analyze two synthetic datasets with objects of known size and anatomical location based on the parameters selected for the study. The validator feature in Nutil confirmed that the results were identical to the ground truth. The datasets, ground truth, and results of Nutil v0.7.0 are shared in the BRAINSPACE repository[84]. The QUINT workflow is shared on EBRAINS (https://ebrains.eu/service/quint), with user documentation[89] and user support available through EBRAINS.

## Data availability

The collection of section images, accompanying metadata, atlas-registration files, QCAlign output, and Nutil output is shared as the BRAINSPACE project via the EBRAINS Knowledge Graph[85]. This EBRAINS dataset[85] includes a data descriptor that details the exact contents of each deposited folder, information on how to download the data, as well as how to cite this data. The source QCAlign and Nutil output (QuickNII only and QuickNII and VisuAlign registration) located on this portal were compiled to create Figs. 2–5 and Supplementary Figs. 2, 3. Gene expression data used in this analysis is deposited on GEO (datasets GSE101144, GSE119215, and GSE119408). Individual input files and scripts to reproduce the analyses conducted in this manuscript are included on our GitHub page[84] (as described in the Code Availability section). Supplementary Data 1 lists all the regions and their Allen Brain Atlas ID that comprise the intermediate hierarchy created for this study. Supplementary Data 2 lists the number of assessments contributing to the mean accuracy/inaccuracy or uncertain scores presented in Fig. 3a and Supplementary Fig. 2c. The output of statistical analyses used to denote significance in the plots are included in Supplementary Data 3-5 (Supplementary Data 3: Fig. 3, Supplementary Data 4: Fig. 4, Supplementary Data 5: Fig. 5). Supplementary Data 6 lists the origin of regions selected, any abbreviations used to denote these regions in figures, and in what analyses these regions were included.

## Code availability

R scripts used to organize, analyze, and complete statistical analyses of the represented data are publicly available on the BRAINSPACE GitHub repository[84]. More information about the use of each script is available on the BRAINSPACE GitHub Wiki page. This wiki page details the purpose of each script, the input data, the output data, and the application of data in the manuscript. The data necessary to implement these scripts are located in the corresponding script folder or can be downloaded from the EBRAINS portal[85] and the GEO (GEO:GSE101144, GEO:GSE119215, GEO:GSE119408)[9,10,12].

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

## Acknowledgements
This study is part of the National Institute on Aging Resilience-AD program and is supported through the NIA parent grant Systems Genetics Analysis of Resilience to Alzheimer's disease: R01AG057914 and supplements R01AG057914-02S1 and R01AG057914-03S1 awarded to Dr. Catherine Kaczorowski, The Jackson Laboratory. The software tools, developed by the Neural Systems Laboratory, University of Oslo, Norway, were funded by EU Horizon 2020, Specific Grant Agreement No. 945539 (Human Brain Project SGA3) awarded to Dr. Jan G. Bjaalie. Workflow optimization for brain-wide spatial analysis to identify regional and cell-type correlates of resilience to Alzheimer's in the AD-BXD mouse population (BRAINSPACE project) received support from the HBP Voucher Programme call 2019 (ID 66) awarded to Dr. Catherine Kaczorowski and Dr. Maja Puchades. We thank the EBRAINS data curation team for their valuable assistance with data management and metadata curation related to the sharing of data through the EBRAINS Knowledge Graph. We also thank Dr. Stephanie Boas for their creative inspiration for figure generation.

## Author contributions
Writing of manuscript: B.G. and S.C.Y. Revision of manuscript: B.G., S.C.Y., N.H., M.T., K.M.S.O., T.M., I.B., H.K., T.B.L., M.A.P, J.G.B., and C.C.K. Processing, registration, and segmentation of brain images: B.G., N.H., E.M. Curation of registration and segmentation of brains: S.C.Y. and M.A.P. Generation of results for QUINT workflow validation in QCAlign: B.G., S.C.Y., M.T., A.O., T.O., S.S., T.M., I.B., H.K. and U.S. Data analysis (QCAlign output, Nutil output, IHC-RNAseq integration): B.G., S.C.Y., N.H. and M.T. Data interpretation: B.G, S.C.Y, N.H, M.T, C.C.K., K.M.S.O., M.A.P., J.G.B Development of experimental design: C.C.K, J.G.B, M.A.P. Development of QUINT tools (VisuAlign, QCAlign, QNLMask, Nutil): S.C.Y., G.C., N.E.G., T.B.L, M.A.P., J.G.B. All authors reviewed and approved the final manuscript.

## Competing interests
The authors declare no competing interests.
