## [Peer Review File · Communications Biology]

Reviewers' comments:

Reviewer #1 (Remarks to the Author):

This is a tour de force by a rigorous group of scientists - and communicates an important advance using genetically diverse mice cross bred onto an AD model. The authors apply an innovative and updated method to apply high throughout approaches in understanding cell numbers, expression and across strains. This is important because it enables one to understand genetic sources of resilience and vulnerability. The manuscript is well written and the data are strong. I would recommend in the abstract and through the manuscript to communicate very directly and simply about the bottom line of what these methods bring to the table. For example, in the abstract, a reader could get easily lost in understanding the "QUINT workflow" - it needs to be simply defined. Lastly, the authors can help to make sure the manuscript is understandable to all audiences with statements such as "this is important because". Overall, awesome work.

Reviewer #2 (Remarks to the Author):

The authors of this manuscript adeptly expanded the capabilities of the previously published QUINT tool/workflow, enhancing its precision in cell type and pathological quantifications. They have recognized and addressed the essential task of filtering and evaluating age-related differences correlated with AD phenotype/pathology. This meticulous approach led to the identification of a distinct subset of genes that, intriguingly, showed no age-related changes in expression but were associated with variations in load within the examined age groups.

A commendable aspect of this work is the authors' transparency about the limitations of both current methods and the enhanced QUINT workflow. Their candid discussion contributes to a balanced and comprehensive understanding of the tool's application in quantifying cell and pathology changes.

Major Comments:

In the results section "Mediation of age reveals differential overrepresentation of Reactome pathways", it would enhance clarity if additional details about the WebGestalt GSEA tool and citation were provided, including the underlying algorithms and calculation methods. This would offer readers a deeper insight into the processes that led to the findings presented in Figure 5.

In discussing significant changes being limited to NeuN in certain regions (notably the hippocampus), it would be enriching to include references corroborating this trend. An expanded explanation or hypothesis regarding the observed trend in the AD-BXD strain would further contribute to the readers' understanding.

Minor Comments:

The absence of a discussion on tau pathology is noticeable. While it might not be a core focus, explaining the omission of phosphorylation of tau markers would contribute to a comprehensive discourse.

The mention of "R scripts" would benefit from additional context. Elaborating on the specific contributions of these scripts, such as input/output processing or the calculation of GSEA scores, and providing links or names of the specific scripts in the repository would enhance accessibility and understanding for the readers.

Reviewer #3 (Remarks to the Author):

Gurdon et al described a study on mapping IHC-stained mouse brain images to reference atlas of the brain in order to understand changes in brain composition that occur with AD pathology. To achieve this goal, they proposed an expansive method of the existing image analysis workflow named QUINT by adding extra functions of brain image acquisition and quality control assessment. They also assessed the combination of the quantified biomarkers from IHC-based image analysis and gene expression changes at adult mice in bulk RNA-seq-based analysis.

The manuscript is extensive, and it is clearly written. The extensive data are shown to lead the conclusion. The methodologies used in this manuscript is well described and it seems to be appropriate, however, I think there are some shortages of explanation. I have also concern about the major conclusion from the data shown.

1) If the authors claim the function named 'VisuAlign' as one of the main achievements of this work, they should explain how it was implemented and/or how it works to help readers understand the results in Figure 1 and 2. The authors just mentioned that it was "nonlinear adjustment", but it is not enough to understand the advantage of the addition of VisuAlign.

2) There seems no summary or detailed explanation of bulk RNA-seq-based gene expression analysis. The data used in this study were derived from the author's previous work, though, summary of the data, such as population, number of replicates and statistical method used, should be indicated in the manuscript.

3) Differently expressed genes from bulk RNA-seq analysis were confirmed by quantified marker values (NeuN, GFAP, Iba1 and amyloid-beta) using the proposed IHC image analysis-based methods in Figure 4. Genes selected by this method could be promising but were there any gene related to AD pathology that was significantly changed but were not selected? It should be evaluated to indicate the capability of the proposed method.

Title: Detecting the effect of genetic diversity on brain composition in an Alzheimer's disease mouse model

Authors: Brianna Gurdon^{1,2#}, Sharon C. Yates^{3#}, Gergely Csucs³, Nicolaas E. Groeneboom³, Niran Hadad¹, Maria Telpoukhovskaia¹, Andrew Ouellette^{1,2}, Tionna Ouellette^{1,4}, Kristen M. S. O'Connell^{1,2,4}, Surjeet Singh¹, Thomas J. Murdy¹, Erin Merchant¹, Ingvild Bjerke³, Heidi Kleven³, Ulrike Schlegel³, Trygve B. Leergaard³, Maja A. Puchades³, Jan G. Bjaalie^{3*}, and Catherine C. Kaczorowski^{1,2,4,5*}

#, contributed equally

*, co-corresponding authors

We thank the reviewers for their time, careful review, and insightful comments. The reviewers agreed that the manuscript is an important advancement in the field and that using the genetically diverse AD-BXD panel to model Alzheimer's disease (AD) in mice and applying an innovative and updated QUNIT workflow results in enhanced cell and pathological quantification. Our revised submission directly and substantially addresses their critiques, which were centered on the inclusion of additional descriptions and enhanced clarity of methodological approaches including the use of VisuAlign, instructions for data and code use, and clarity summarizing the main findings of the paper.

The reviewer's original comments are pasted in full and were subsequently broken down according to the main points. Our response to each comment is included below. Line numbers refer to the final location of the revised or added text after all tracked changes are accepted.

Reviewer #1 (Remarks to the Author):

This is a tour de force by a rigorous group of scientists - and communicates an important advance using genetically diverse mice cross bred onto an AD model. The authors apply an innovative and updated method to apply high throughout approaches in understanding cell numbers, expression and across strains. This is important because it enables one to understand genetic sources of resilience and vulnerability. The manuscript is well written and the data are strong. I would recommend in the abstract and through the manuscript to communicate very directly and simply about the bottom line of what these methods bring to the table. For example, in the abstract, a reader could get easily lost in understanding the "QUINT workflow" - it needs to be simply defined. Lastly, the authors can help to make sure the manuscript is understandable to all audiences with statements such as "this is important because". Overall, awesome work.

R1.1: I would recommend in the abstract and through the manuscript to communicate very directly and simply about the bottom line of what these methods bring to the table.

Authors' Response: In response to this suggestion and comment R1.3 additional statements throughout the manuscript were added to provide concise descriptions of the tools used and analyses run to emphasize their importance and overall benefit. These additions are listed under R1.3.

R1.2: In the abstract, a reader could get easily lost in understanding the "QUINT workflow" - it needs to be simply defined.

Authors' Response: A simple definition of the QUINT workflow as a suite of software designed to support atlas-based quantification was added to the abstract to enhance the clarity of the purpose of the QUINT workflow.

Text added to the abstract: "We utilized the analytical QUINT workflow- an established suite of software designed to support atlas-based quantification, which we expanded to deliver a highly

effective method for registering and quantifying cell and pathology changes in diverse disease models.”

R1.3: The authors can help to make sure the manuscript is understandable to all audiences with statements such as "this is important because".

Authors' Response: To address this comment and present our data in an approachable fashion we added summary statements to clarify main findings and summarize results sections. Such sentences are found in the following sections of the manuscript:

Section: New functionality added to the QUINT workflow supports high-throughput analysis of diverse AD-BXD strains, lines: 121-125: “Here we demonstrate the effectiveness of the expanded QUINT workflow to quantify diverse cellular and pathological features in heterogenous brain tissue of AD-BXD mice (Figure 1b) by quantifying all nuclei (thionine), neurons (NeuN), microglia (Iba1), astrocytes (GFAP) and amyloid-beta (AB1-42) in customized regions compiled from CCFv3 regions (Figure 1c-d).”

Section: Nonlinear adjustment increases regional registration accuracy and impacts cell and pathology load estimates, lines: 188-191: “In conclusion, the capability to perform nonlinear adjustments to QuickNII atlas-registration in VisuAlign is crucial because it significantly improves regional registration accuracy, particularly in complex regions like the hippocampus, leading to more reliable and accurate cell and pathology load estimates.”

Section: Individual AD-BXD strains exhibit variation in region neuronal load, lines: 225-228: “Understanding that genetics strongly contribute to variation in symptom onset and susceptibility to AD in both humans^{41,42} and AD-BXD mice^{9,43,44}, here we have highlighted the translatable potential to investigate the influence of genetic background on the presentation of neurodegeneration in animals carrying the 5XFAD transgene.”

Section: Integration of paired IHC and bulk RNA sequencing data reveals cell load is a confounding factor in age-by-gene expression correlations among AD-BXDs, lines: 288-298: “In conclusion, our analysis demonstrates that variations in cell and amyloid-beta load can significantly affect the interpretation of age-by-gene expression correlations. This highlights the importance of considering cell composition as a potentially confounding factor in gene expression analyses, particularly in studies involving age-related diseases like AD. By separating age-dependent and age-independent gene correlates, we could better distinguish between genes whose expression changes directly with age and AD pathogenesis versus those whose expression changes are driven by age-related alterations in cell populations. This distinction helps inform whether candidate gene expression (e.g., overexpress or knockdown gene expression) or cell/amyloid-beta composition (e.g., target the maintenance of a cell type’s load) should be targeted. This analysis also provides valuable insights into the complex interplay between aging, cell composition, and gene expression in AD.”

Reviewer #2 (Remarks to the Author):

The authors of this manuscript adeptly expanded the capabilities of the previously published QUINT tool/workflow, enhancing its precision in cell type and pathological quantifications. They have recognized and addressed the essential task of filtering and evaluating age-related differences correlated with AD phenotype/pathology. This meticulous approach led to the identification of a distinct subset of genes that, intriguingly, showed no age-related changes in expression but were associated with variations in load within the examined age groups.

A commendable aspect of this work is the authors' transparency about the limitations of both current methods and the enhanced QUINT workflow. Their candid discussion

contributes to a balanced and comprehensive understanding of the tool's application in quantifying cell and pathology changes.

Major Comments:

In the results section “Mediation of age reveals differential overrepresentation of Reactome pathways”, it would enhance clarity if additional details about the WebGestalt GSEA tool and citation were provided, including the underlying algorithms and calculation methods. This would offer readers a deeper insight into the processes that led to the findings presented in Figure 5.

In discussing significant changes being limited to NeuN in certain regions (notably the hippocampus), it would be enriching to include references corroborating this trend. An expanded explanation or hypothesis regarding the observed trend in the AD-BXD strain would further contribute to the readers' understanding.

Minor Comments:

The absence of a discussion on tau pathology is noticeable. While it might not be a core focus, explaining the omission of phosphorylation of tau markers would contribute to a comprehensive discourse.

The mention of “R scripts” would benefit from additional context. Elaborating on the specific contributions of these scripts, such as input/output processing or the calculation of GSEA scores, and providing links or names of the specific scripts in the repository would enhance accessibility and understanding for the readers.

R2.1: In the results section “Mediation of age reveals differential overrepresentation of Reactome pathways”, it would enhance clarity if additional details about the WebGestalt GSEA tool and citation were provided, including the underlying algorithms and calculation methods.

Authors' Response: Additional detail pertaining to the calculation methods of WebGestalt is now included via the addition of the following citations for GSEA overall (citation 50) and WebGestalt (citations 51 & 52). Screenshots of the exact parameters used in the current web version of WebGestalt are now also provided via the BRAINSPACE GitHub page.

49. Subramanian, A. et al. Gene set enrichment analysis: A knowledge-based approach for interpreting genome-wide expression profiles. *Proceedings of the National Academy of Sciences* **102**, 15545–15550 (2005).
50. Zhang, B., Kirov, S. & Snoddy, J. WebGestalt: an integrated system for exploring gene sets in various biological contexts. *Nucleic Acids Res* **33**, W741-748 (2005).
51. Liao, Y., Wang, J., Jaehnig, E. J., Shi, Z. & Zhang, B. WebGestalt 2019: gene set analysis toolkit with revamped UIs and APIs. *Nucleic Acids Research* **47**, W199–W205 (2019).

Screenshot added to GitHub documentation:

Basic parameters

Organism of Interest ⓘ

Method of Interest ⓘ

Functional Database ⓘ

Gene List

Select Gene ID Type ⓘ

Upload Gene List ⓘ

OR

Information entered:
Ensembl gene IDs and correlation coefficients (R values) from age-adj and non-age-adj correlation output per stain

ENSMUSG00000000001 0.225619859

ENSMUSG00000000028 0.072481075

ENSMUSG00000000037 0.150502777

ENSMUSG00000000056 -0.090495635

ENSMUSG00000000058 -0.020456500

Advanced parameters ⌵

minimum number of genes for a category ⓘ

Maximum number of genes for a category ⓘ

Significance Level ⓘ FDR TOP

Number of Permutations ⓘ

p ⓘ

Collapse Method ⓘ

Number of categories expected from set cover ⓘ

Number of categories visualized in the report ⓘ

Color in DAG ⓘ Continuous Binary

The following text detailing the input and output parameters and the goal of the analysis was added in the first paragraph of the **Mediation of age reveals differential overrepresentation of Reactome pathways** section.

Manuscript Lines 301-308 Edited/Added: “Next, using the correlation coefficients in Figures 5a and 5b, gene set enrichment analysis (GSEA)⁵⁰ was performed via WebGestalt^{51,52} to extract biological insights from genes of interest and ultimately identify pathways biased by individual differences in cell and amyloid-beta load (Figure 6). The gene Ensembl IDs and associated correlation coefficients calculated via the age-dependent or age-independent multilevel correlations discussed above were input into WebGestalt. The output normalized enrichment scores adjusted for multiple test corrections (FDR) were evaluated to determine whether gene sets for biological pathways are enriched among the positive and/or negative multilevel correlations.”

R2.2: In discussing significant changes being limited to NeuN in certain regions (notably the hippocampus), it would be enriching to include references corroborating this trend.

Authors' Response: The authors agreed with the reviewer's remark and added an additional comparison of results between previous studies characterizing the 5XFAD model and our study to the discussion section. Furthermore, the Kaczorowski lab is actively working to expand upon the findings of this paper by including a greater representation of AD-BXD strains, both sexes, and nontransgenic animals. Preliminary data from this larger dataset is aligned with published findings that 5XFAD-related neurodegeneration occurs at our adult (6m old) timepoint and is exacerbated with age. This dataset that is powered to evaluate differences in stain load

between Ntgs and 5XFADs will be utilized to rigorously address mechanisms of neurodegeneration and resilience to AD.

Manuscript Lines 387-395 Added: “In contrast to the initial qualitative³⁷ and later quantitative^{39,65,66} studies that identified significant neuron loss in cortical layer V and subiculum starting at 6m in 5XFADs compared to controls, our study, due to the nature of the dataset, could only detect age-related neurodegeneration in 5XFADs. While these studies did not measure neurodegeneration in the hippocampus overall, more recent studies quantified a decrease in NeuN protein in the hippocampus by 8m⁶⁸, 10m⁶⁹, and 12m⁶⁷ in 5XFAD mice compared to Ntgs. The differences in age, sex, and genetic background as well as analytical differences amongst studies (e.g. regions compared) may explain the discrepancy in the detection of neurodegeneration in these studies and ours.”

R2.3: An expanded explanation or hypothesis regarding the observed trend [neurodegeneration in the hippocampus] in the AD-BXD strain would further contribute to the readers’ understanding.

Authors’ Response: The classification of AD-BXD strains is a focus in the Kaczorowski lab and we concur that the detection of this trend of variation of hippocampal neurodegeneration is an important finding to expand upon. Additional information regarding the potential impact of genetic background on the extent of neurodegeneration in the characterized AD-BXD strains as well as a hypothesis of what may be driving this variation are now included in the discussion of the manuscript.

Manuscript Lines 395-410 Added: “Since we studied the effects of the 5XFAD transgene using the diverse AD-BXD panel, we were uniquely positioned to detect variation in NeuN load among AD-BXD strains. We demonstrated a trend that certain AD-BXD strains exhibit a decrease in NeuN load from 6m to 14m while other strains do not. We believe that this NeuN variation is not due to varying APP expression levels as we have previously measured transgene expression via quantification of human APP expression and endogenous mouse APP levels and found no significant differences among the panel of strains evaluated⁹. Instead, we hypothesize that genetic differences amongst the AD-BXD strains may influence how each strain copes with the effects of the 5XFAD transgene and aging^{2,9,10,44}. Ultimately, AD-BXD strains can be classified into general AD subtypes⁷⁰ or stratified as resilient or susceptible to AD pathology¹¹: with resilient strains potentially mitigating neuron loss in response to neuroinflammation and pathology accumulation or staving off severe pathology accumulation altogether. Moreover, the AD-BXD panel has proven to be a robust population for genetic mapping of behavioral traits^{9-12,71,72}, and ongoing experiments performing quantitative trait loci mapping aim to elucidate genetic factors responsible for variation in heritable regional cell and amyloid-beta load^{73,74}.”

R2.4: The absence of a discussion on tau pathology is noticeable. While it might not be a core focus, explaining the omission of phosphorylation of tau markers would contribute to a comprehensive discourse.

Authors’ Response: We acknowledge that the lack of consideration of tau is a limitation of the manuscript and the study overall. There is now a growing body of literature exploring the presence of tau pathology in 5XFAD animals¹⁻⁸ and the Kaczorowski lab is similarly interested in investigating the extent of tau presence in the AD-BXD panel. We have initiated this process and contracted NeuroScience Associates to perform a pilot study conducting IHC staining for phosphorylated tau (AT8, 9G3, and pSer396 antibodies); however, this experiment resulted in very diffuse AT8 and 9G3 staining not amenable to segmentation and variable pSer369 staining (unpublished). From that pilot study, we deduced that the AD-BXD strains investigated do not develop overt pathological tau tangles, so we chose not to pursue the characterization of tau as

part of this study. Further troubleshooting guided by the recent literature confirming pathological tau in 5XFAD mice is required. Moreover, we are currently engaged in additional molecular characterization of abnormal tau in the AD-BXD panel. We have evaluated various phosphorylated tau species (AT8, 12E8, 9G3, and TOMA2) in the AD-BXD founders and found that there are differences between some ptau species in B6xB6:5XFAD and B6:5XFADxDBA/2J mice (we did not include other AD-BXD strains) (unpublished).

1. Griñán-Ferré, C. *et al.* Epigenetic mechanisms underlying cognitive impairment and Alzheimer disease hallmarks in 5XFAD mice. *Aging* **8**, 664–684 (2016).
2. Mattsson-Carlgren, N. *et al.* A β deposition is associated with increases in soluble and phosphorylated tau that precede a positive Tau PET in Alzheimer's disease. *Science Advances* **6**, eaaz2387 (2020).
3. Claeysen, S., Giannoni, P. & Ismeurt, C. The 5xFAD mouse model of Alzheimer's disease. in *The Neuroscience of Dementia* vol. 1 207–221 (Academic Press, 2020).
4. Tohda, C., Urano, T., Umezaki, M., Nemere, I. & Kuboyama, T. Diosgenin is an exogenous activator of 1,25D3-MARRS/Pdia3/ERp57 and improves Alzheimer's disease pathologies in 5XFAD mice. *Sci Rep* **2**, 535 (2012).
5. Maarouf, C. L. *et al.* Molecular Differences and Similarities Between Alzheimer's Disease and the 5XFAD Transgenic Mouse Model of Amyloidosis. *Biochem Insights* **6**, 1–10 (2013).
6. Choi, H.-J. *et al.* Donepezil ameliorates A β pathology but not tau pathology in 5xFAD mice. *Molecular Brain* **15**, 63 (2022).
7. Cho, H.-J., Sharma, A. K., Zhang, Y., Gross, M. L. & Mirica, L. M. A Multifunctional Chemical Agent as an Attenuator of Amyloid Burden and Neuroinflammation in Alzheimer's Disease. *ACS Chem Neurosci* **11**, 1471–1481 (2020).
8. Shukla, V. *et al.* A truncated peptide from p35, a Cdk5 activator, prevents Alzheimer's disease phenotypes in model mice. *The FASEB Journal* **27**, 174–186 (2013).

We also recognize that novel mouse models integrating tau pathology on the 5XFAD background are now being developed to investigate the interplay between pathological levels of amyloid-beta and tau^{1,2}. This model could be a great resource for investigating the interaction of human amyloid and tau pathology. Future investigation of changes in cell and pathology composition in these mice may further the study of AD and the road to developing therapeutic solutions for patients.

1. Farfara, D. *et al.* Physiological expression of mutated TAU impaired astrocyte activity and exacerbates β -amyloid pathology in 5xFAD mice. *J. Neuroinflammation* **20**, 174 (2023).
2. Barendrecht, S. *et al.* A novel human tau knock-in mouse model reveals interaction of Abeta and human tau under progressing cerebral amyloidosis in 5xFAD mice. *Alzheimers Res. Ther.* **15**, 16 (2023).

The previous omission of tau pathology in the AD-BXD panel is now addressed with a paragraph in the discussion regarding the current research investigating tau in 5XFAD animals, the transcriptional relevance of the AD-BXD panel to human AD, and the mostly unexplored widespread investigation of tau pathology in the AD-BXD panel.

Manuscript Lines 448-461 Added: “A limitation of our study is the lack of consideration of tau pathology. Initially, the 5XFAD transgene was not thought to induce significant tau tangle pathology as no human tau transgene is expressed in this mouse line³⁷; therefore, the 5XFAD model was prominently used as a model to investigate A β -associated neurodegeneration and neuron loss⁶⁶. More recent investigations have reported the presence of varying pathogenic phospho-tau proteins at different tau residue sites in the brains of 5XFAD animals^{68,78–82}. We have verified that AD-BXD animals exhibit strong amyloid and neuroinflammatory responses with age and AD, display a high level of concordance with both familial and sporadic forms of

human AD at the molecular and behavioral level⁹, and that female AD-BXD mice exhibit high translational alignment and conserved cell-type-specific signatures of resilience to AD with human AD cohorts¹¹, but technical limitations have impeded our ability to explore the presence of tau epitopes across the AD-BXD panel. These consistent findings characterizing tau in 5XFAD brains warrant the future investigation of phosphorylated tau in AD-BXD strains. Like the variation in NeuN we described here, we predict that strain-specific variation in tau may also be present in this panel.”

R2.5: The mention of “R scripts” would benefit from additional context. Elaborating on the specific contributions of these scripts, such as input/output processing or the calculation of GSEA scores, and providing links or names of the specific scripts in the repository would enhance accessibility and understanding for the readers.

Authors' Response: All of the scripts used to perform the analyses and data visualization presented in this paper are included on the BRAINSPACE Github page¹. We agree with the reviewer that the “R scripts” could benefit from more context, and have thereby added a Wiki page to the BRAINSPACE Github page that describes the purpose, input, and output of each script. It also describes where the relevant data is presented in the manuscript, allowing a reader to reproduce the findings presented in this manuscript using the data deposited on the EBRAINS portal². We decided to keep this information attached to the R scripts in GitHub to promote continuity of retrieving and analyzing data.

1. BRAINSPACE. <https://github.com/Neural-Systems-at-UIO/BRAINSPACE> (2023).
2. Gurdon, B. *et al.* Investigating cellular diversity in a novel Alzheimer's disease mouse model using the optimized QUINT workflow. *EBRAINS* <https://doi.org/10.25493/SZ0M-EE6> (2023).

Reviewer #3 (Remarks to the Author):

Gurdon et al described a study on mapping IHC-stained mouse brain images to reference atlas of the brain in order to understand changes in brain composition that occur with AD pathology. To achieve this goal, they proposed an expansive method of the existing image analysis workflow named QUINT by adding extra functions of brain image acquisition and quality control assessment. They also assessed the combination of the quantified biomarkers from IHC-based image analysis and gene expression changes at adult mice in bulk RNA-seq-based analysis.

The manuscript is extensive, and it is clearly written. The extensive data are shown to lead the conclusion. The methodologies used in this manuscript is well described and it seems to be appropriate, however, I think there are some shortages of explanation. I have also concern about the major conclusion from the data shown.

1) If the authors claim the function named ‘VisuAlign’ as one of the main achievements of this work, they should explain how it was implemented and/or how it works to help readers understand the results in Figure 1 and 2. The authors just mentioned that it was “nonlinear adjustment”, but it is not enough to understand the advantage of the addition of VisuAlign.

2) There seems no summary or detailed explanation of bulk RNA-seq-based gene expression analysis. The data used in this study were derived from the author’s previous work, though, summary of the data, such as population, number of replicates and statistical method used, should be indicated in the manuscript.

3) Differently expressed genes from bulk RNA-seq analysis were confirmed by quantified marker values (NeuN, GFAP, Iba1 and amyloid-beta) using the proposed IHC image

analysis-based methods in Figure 4. Genes selected by this method could be promising but were there any gene related to AD pathology that was significantly changed but were not selected? It should be evaluated to indicate the capability of the proposed method.

R3.1: If the authors claim the function named 'VisuAlign' as one of the main achievements of this work, they should explain how it was implemented and/or how it works to help readers understand the results in Figure 1 and 2. The authors just mentioned that it was "nonlinear adjustment", but it is not enough to understand the advantage of the addition of VisuAlign.

Authors' response: The creation of VisuAlign is a major addition to the QUINT workflow. We agree with the reviewer's comment and now provide further explanation about the tool's functionality and divided the former Figure 2 into 2 figures to emphasize the importance of VisuAlign. The new Figure 2 provides additional details about the registration and verification process in the QUINT workflow using VisuAlign and QCAlign. This includes a snapshot of the user experience within VisuAlign and an example of the registration quality that can be achieved with QuickNII only (linear registration) and with QuickNII and VisuAlign in succession (linear registration with nonlinear refinement). The figure also demonstrates the use of QCAlign to verify the registration achieved before refinement and after nonlinear refinement, highlighting the importance of the ability to refine and improve the atlas-registration with the new VisuAlign tool.

Additional text was added to the following sections of the manuscript:

Section: Nonlinear adjustment increases regional registration accuracy and impacts cell and pathology load estimates, lines: 167-172: "VisuAlign provides users with an accessible graphical user interface (GUI) where they can systematically make nonlinear adjustments to regional boundaries of the atlas-plates from the Allen Mouse Brain Atlas as set in QuickNII, to more accurately reflect the structural composition of the experimental section (Figure 2a). This process involves identifying mismatches between the atlas-plate and the underlying experimental section and manually positioning and dragging anchor points on the atlas-plate to their correct position on the section."

Methods Section: QUINT workflow development, lines: 567-571: "VisuAlign is used to apply in-plane nonlinear adjustments to the atlas-plates established in QuickNII to achieve the best fit over the sections. Nonlinear adjustments are made by identifying mismatches between the atlas-plate and the underlying section in the VisuAlign GUI, manually positioning anchor points on the atlas-plate, and dragging the points to their correct position on the section."

R3.2: There seems no summary or detailed explanation of bulk RNA-seq-based gene expression analysis. The data used in this study were derived from the author's previous work, though, summary of the data, such as population, number of replicates and statistical method used, should be indicated in the manuscript.

Authors' response: Thank you for your attention to detail regarding the explanation of methods in this manuscript. We have acknowledged your concern by adding an additional "Bulk RNA Sequencing" methods section. This section states 1. the population description of the RNAseq data collected in Neuner et al., 2019 and how the 34 samples we included in our analyses were chosen (also now included as an additional column in Supplemental Table 1), 2. a brief description of how the tissue for this RNAseq analysis was collected and processed, and 3. a synopsis of how the gene expression data underwent filtering and the R packages used to analyze the data.

This information was conveyed in the following lines added to the manuscript, lines: 708-720: "In brief, RNA was isolated via a QIAcube using the RNeasy mini kit (QIAGEN) and

treated with DNase to remove contaminating DNA. RNA quality was confirmed using a BioAnalyzer (Agilent Technologies). All samples had RNA Integrity Numbers (RIN values) > 8.0. Sequencing libraries were prepared from 1 µg RNA with the Truseq Stranded mRNA Sample Preparation Kit (Illumina Inc). Final PCR-enriched fragments were validated on a 2200 TapeStation Instrument using the D1000 ScreenTape (Agilent Technologies) and quantified by qPCR using a Universal Library Quantification Kit (Kapa Biosystems) on the QuantStudio 6 Flex (ThermoFisher Scientific). Final library pools were sequenced by 75bp paired-end sequencing on a HiSeq2500 (Illumina Inc). Because both C57BL/6J and DBA/2J alleles segregate within our panel, the GBRS/EMASE pipeline⁸⁷ developed by the Churchill group at The Jackson Laboratory was used in order to align reads to a diploid transcriptome. An expectation maximization algorithm was used in order to align reads to the correct allele. This method allows for the quantification of both total reads assigned to a gene and the number of reads assigned to either the B or D allele.”

R3.3: Differently expressed genes from bulk RNA-seq analysis were confirmed by quantified marker values (NeuN, GFAP, Iba1, and amyloid-beta) using the proposed IHC image analysis-based methods in Figure 4. Genes selected by this method could be promising but were there any gene related to AD pathology that was significantly changed but were not selected? It should be evaluated to indicate the capability of the proposed method.

Authors' response: The genes selected to be represented in Figure 4d are only a small snapshot of the gene expression and cell composition relationships found as a result of the multilevel correlation. While no significant age-adjusted gene correlated with AB1-42 passed our FDR correction, several non-age-adjusted genes were significantly correlated with AB1-42 load after FDR correction and these genes could have also been displayed in additional figure panels. Similar to the pattern of gene expression and Iba1 load represented in Figure 4d i, the top non-age-adjusted genes positively significantly correlated with AB1-42 load after FDR correction, including C4b (most highly correlated gene, FDR-corrected p-value: 3.19E-05), exhibit an increase in gene expression and pathology load with age. We chose to only represent the top correlates for one stain because the pattern of increased stain load with age as well as a change in gene expression with age (positive or negative depending of the direction of correlation) was consistent among stains prior to age adjustment (see attached extended figure 4d). Likewise, the pattern of increased cell and amyloid-beta load in age but no change in gene expression was seen among the few FDR significant gene correlated after age-adjustment. The output from the multilevel correlation grouped by correlation adjustment is represented in Supplemental Table 8.

The consistency of the relationship between increased GFAP, Iba1, and AB1-42 load and gene expression among FDR-significant correlations was previously referenced in the sentence: “This trend of increased load matched by a change in gene expression between 6m and 14m was unique to the most highly correlated genes prior to age adjustment.”; however, we acknowledge that we can expand upon this concept.

Additional information regarding the pattern of top FDR-significant genes correlates with GFAP, Iba1, and NeuN load before and after age adjustment is **now included in the manuscript, lines: 280-288**: “We found that the relationship between gene expression and GFAP, Iba1, and AB1-42 load is consistent among the topmost significant correlated genes following FDR correction for each adjustment method (not age-adjusted or age-adjusted). Genes exclusively significant prior to age adjustment exhibit an age-related increase in gene expression that mirrors the age-related increase in hippocampal formation AB1-42, GFAP, and Iba1 load. Similarly, when evaluating the most significant correlated genes with GFAP, AB1-42, or Iba1 load exclusively after age adjustment, we saw a consistent pattern of increased regional stain load with age without age-related increases in gene expression.”

Extended Reviewer Figure 4d.) Individual relationship between gene expression and load with age for the top age-dependent and independently correlated genes with Iba1 and AB1-42.

i. *Galnt6* was exclusively significantly correlated with Iba1 without age adjustment. An increase in Iba1 load and *Galnt6* expression occurs between 6m and 14m. A positive relationship between Iba1 load and *Galnt6* expression exists across both age groups as well as within each age group. ii. *Tmem39a* was exclusively significantly correlated with Iba1 after age adjustment. An increase in Iba1 load but not in *Tmem39a* expression occurs between 6m and 14m. A weak relationship between Iba1 load and *Tmem39a* expression exists across both age groups, but separate age-specific correlations with load and gene expression exist. iii. *C4b* was the most significantly positively correlated gene with AB1-42 load without age adjustment. An increase in

AB1-42 load and *C4b* expression occurs between 6m and 14m. A positive relationship between AB1-42 load and *C4b* expression exists across both age groups as well as within each age group. *Rtn1* was the most significantly negatively correlated gene with AB1-42 load without age adjustment. A negative relationship between AB1-42 load and *Rtn1* expression exists across both age groups as well as within each age group. 5XFAD mice only, 6m: n=17, 14m: n=20.

Author Note: Minor additional changes to improve consistency and clarity while promoting adherence to the suggested word limit are also included in the markup document. Figures have also been created or adapted to account for reviewers' requests.

The following changes to the figures have been made:

1. Creation of new figure 2: To emphasize the impact of adding VisuAlign and QCAIign to the QUINT workflow we have created a new figure representing the graphic user interface and use of each of these tools. This figure visually highlights the manual adjustment of Allen Mouse Brain Atlas regions required to be completed in VisuAlign (Figure 2a) to improve regional registration accuracy as measured in QCAIign (Figure 2b).

Figure 2. VisuAlign and QCAIign were used to refine and verify the regional atlas-registration achieved in the QUINT workflow. a). VisuAlign GUI displaying one thionine section with nonlinear refinements applied to achieve an improved match of the atlas delineations over the section. CCFv3 regional borders are overlaid on the section with the position of the borders manipulated using anchor points. The lines indicate the start position of the points prior to nonlinear refinement, with the black markers denoting their final position after nonlinear refinement. i). Inset displaying the atlas-registration achieved by linear registration using QuickNII. The dentate gyrus cell layers are incorrectly positioned over the section. ii). Inset displaying the atlas-registration achieved using QuickNII and VisuAlign. The positioning of the dentate gyrus cell layers has been adjusted to match the cell layers in the section. b). QCAIign GUI displaying one thionine section with a grid of systematic random sampling points overlaid. Grid points are marked up as registered accurately (+) or inaccurately (-) based on the region name, which is displayed in the GUI by hovering over a point (region name shown for point indicated with the arrow). iii). Inset displaying the quality of the atlas-registration achieved by linear registration with QuickNII only (87% accurate for the inset) (iii). Inset displaying the quality of the atlas-registration achieved by registration with QuickNII and VisuAlign (with nonlinear refinement) (100% accurate for inset iv).

- Current Figure 3a was adjusted to comply with the Nature portfolio requirements of showing individual data points for any mean points that have $n \leq 10$ data points contributing to that mean. Previously, several regions had less than 10 data points contributing to the mean point, and individual points were not represented, only the mean \pm SEM was shown. This plot has now been updated to show the individual data points contributing to the mean. The QCAAlign accuracy score calculated following QuickNII registration alone (Navy, triangles) vs accuracy scores calculated after QuickNII and VisuAlign registration (green circles) are now clearly demarcated. The mean accuracy score per region and registration method are indicated with a dark shape (triangle or circle) \pm SEM. The individual points contributing to that mean score are located beneath the mean point and are opaque in color.

Figure 3
a

Figure 3. QCAAlign verification of regional atlas-registration at the selected intermediate hierarchy level. a.) Mean accuracy scores per intermediate hierarchy region after QuickNII registration alone (navy triangles) or after QuickNII and VisuAlign registration (green circles). Two raters scored the same 5 randomly selected brains after QuickNII registration alone, max $n=10$ per region (Raters: $n=2$ per brain). Up to 10 raters scored the same 5 randomly selected brains after QuickNII and VisuAlign registration, max $n=36$ per region (Raters: $n=6-10$ per brain). Dark shapes represent the mean score across raters per region for 5 brains \pm SEM, with the opaque shapes representing the individual assessment contributing to each mean calculation. b.) The impact of VisuAlign refinement on regional stain load (%-stain-positive coverage/per region area) was measured by calculating the difference in load following Nutil quantification after each method (regional (QuickNII + VisuAlign output (%)) - regional (QuickNII output (%)) = regional load difference (%)). Dots represent mean regional load difference \pm SEM for all 5XFAD animals at 6m and 14m (6m: $n=17$, 14m: $n=20$).

- Like the changes made in Figure 3a described above, the individual data points contributing to the displayed mean \pm SEM are now displayed on the plot representing the regional mean uncertainty score in Supplemental Figure 2c. The QCAAlign uncertainty score calculated following QuickNII registration alone (Navy, triangles) vs uncertainty scores calculated after QuickNII and VisuAlign registration (green circles) are now clearly demarcated. The mean uncertainty score per region and registration method are indicated with a dark shape (triangle or circle) \pm SEM. The individual points contributing to that mean score are located beneath the mean point and are opaque in color.

Supplemental Figure 2: QCAAlign scores achieved via quality control assessment of intermediate hierarchy regions. a.) Heatmap of regional accuracy scores per rater per brain. b.) Heatmap of regional uncertainty scores per rater per brain. Gray regions were not represented in the brain series and/or did not receive QCAAlign scores for the measure. c.) Mean uncertainty scores per intermediate hierarchy region after QuickNII registration alone (navy triangles) or after QuickNII and VisuAlign registration (green circles). Two raters scored the same 5 randomly selected brains after QuickNII registration alone, max n=10 per region (Raters: n= 2 per brain). Up to 10 raters scored the same 5 randomly selected brains after QuickNII and VisuAlign registration, max n=36 per region (Raters: n= 6-10 per brain). Dark shapes represent the mean score across raters per region for 5 brains +/-SEM, with the opaque shapes representing the individual assessments contributing to each mean calculation.

REVIEWERS' COMMENTS:

Reviewer #2 (Remarks to the Author):

I am satisfied with the changes.

Reviewer #3 (Remarks to the Author):

The authors have addressed well to my concerns I raised in the previous review, so I'd suggest the acceptance.